# *Drosophila* SWR1 and NuA4 complexes are defined by DOMINO isoforms

**Alessandro Scacchetti[1], Tamas Schauer[2], Alexander Reim[3], Zivkos Apostolou[1], Aline Campos Sparr[1], Silke Krause[1], Patrick Heun[4], Michael Wierer[3], Peter B Becker[1]\***

[1]Molecular Biology Division, Biomedical Center, Ludwig-Maximilians-University, Munich, Germany; [2]Bioinformatics Unit, Biomedical Center, Ludwig-Maximilians-University, Munich, Germany; [3]Department of Proteomics and Signal Transduction, Max Planck Institute of Biochemistry, Munich, Germany; [4]Wellcome Trust Centre for Cell Biology and Institute of Cell Biology, School of Biological Sciences, The University of Edinburgh, Edinburgh, United Kingdom

**Abstract** Histone acetylation and deposition of H2A.Z variant are integral aspects of active transcription. In *Drosophila*, the single DOMINO chromatin regulator complex is thought to combine both activities *via* an unknown mechanism. Here we show that alternative isoforms of the DOMINO nucleosome remodeling ATPase, DOM-A and DOM-B, directly specify two distinct multi-subunit complexes. Both complexes are necessary for transcriptional regulation but through different mechanisms. The DOM-B complex incorporates H2A.V (the fly ortholog of H2A.Z) genome-wide in an ATP-dependent manner, like the yeast SWR1 complex. The DOM-A complex, instead, functions as an ATP-independent histone acetyltransferase complex similar to the yeast NuA4, targeting lysine 12 of histone H4. Our work provides an instructive example of how different evolutionary strategies lead to similar functional separation. In yeast and humans, nucleosome remodeling and histone acetyltransferase complexes originate from gene duplication and paralog specification. *Drosophila* generates the same diversity by alternative splicing of a single gene.

**\*For correspondence:**
pbecker@bmc.med.lmu.de

**Competing interests:** The authors declare that no competing interests exist.

## Introduction

Nucleosomes, the fundamental units of chromatin, are inherently stable and organized in polymeric fibers of variable compactness (*Baldi et al., 2018*; *Erdel and Rippe, 2018*). The dynamic properties of the fiber required for gene regulation are implemented by several broad principles. ATP-dependent nucleosome remodeling factors slide or evict nucleosomes (*Clapier et al., 2017*), chemical modifications of histones create new interaction surfaces (*Bowman and Poirier, 2015*; *Zhao and Garcia, 2015*) and histone variants furnish nucleosomes with special features (*Talbert and Henikoff, 2017*).

The very conserved H2A variant H2A.Z accounts for ~5–10% of the total H2A-type histone pool in vertebrates (*Redon et al., 2002*; *Thatcher and Gorovsky, 1994*) and flies (*Bonnet et al., 2019*). H2A.Z is primarily found at active promoters and enhancers, where it is thought to be important to regulate transcription initiation and early elongation (*Adam et al., 2001*; *Weber et al., 2010*; *Weber et al., 2014*). In *Saccharomyces cerevisiae*, H2A.Z is introduced into chromatin by the SWR1 complex (SWR1.C), a multi-subunit ATP-dependent chromatin remodeler with the INO80-type ATPase SWR1 at its core (*Mizuguchi et al., 2004*; *Ranjan et al., 2013*; *Wang et al., 2018a*; *Willhoft et al., 2018*; *Wu et al., 2005*). In humans, the two SWR1 orthologs EP400 and SRCAP may also be involved in H2A.Z incorporation (*Greenberg et al., 2019*; *Pradhan et al., 2016*).

In *Drosophila melanogaster*, where H2A.Z is named H2A.V (*Baldi and Becker, 2013*; *van Daal and Elgin, 1992*), only one gene codes for a SWR1 ortholog: *domino* (*dom*) (*Ruhf et al., 2001*). The

**eLife digest** Cells contain a large number of proteins that control the activity of genes in response to various signals and changes in their environment. Often these proteins work together in groups called complexes. In the fruit fly *Drosophila melanogaster*, one of these complexes is called DOMINO. The DOMINO complex alters gene activity by interacting with other proteins called histones which influence how the genes are packaged and accessed within the cell.

DOMINO works in two separate ways. First, it can replace certain histones with other variants that regulate genes differently. Second, it can modify histones by adding a chemical marker to them, which alters how they interact with genes. It was not clear how DOMINO can do both of these things and how that is controlled; but it was known that cells can make two different forms of the central component of the complex, called DOM-A and DOM-B, which are both encoded by the same gene.

Scacchetti et al. have now studied fruit flies to understand the activities of these forms. This revealed that they do have different roles and that gene activity in cells changes if either one is lost. The two forms operate as part complexes with different compositions and only DOM-A includes the TIP60 enzyme that is needed to modify histones. As such, it seems that DOM-B primarily replaces histones with variant forms, while DOM-A modifies existing histones. This means that each form has a unique role associated with each of the two known behaviors of this complex.

The presence of two different DOMINO complexes is common to flies and, probably, other insects. Yet, in other living things, such as mammals and yeast, their two roles are carried out by protein complexes originating from two distinct genes. This illustrates a concept called convergent evolution, where different organisms find different solutions for the same problem. As such, these findings provide an insight into the challenges encountered through evolution and the diverse solutions that have developed. They will also help us to understand the ways in which protein activities can adapt to different needs over evolutionary time.

first biochemical characterization revealed the presence of a multi-subunit complex composed of 15 proteins associated with the DOM ATPase (*Kusch et al., 2004*). While many of the interactors identified are orthologous to the yeast SWR1.C subunits, additional interactors were found. Surprisingly, they showed similarity to components of a distinct yeast complex, the Nucleosome Acetyltransferase of H4 (NuA4.C) (*Kusch et al., 2004*). NuA4.C is a histone acetyltransferase (HAT) complex with the histone H4 N-terminal domain as a primary target (*Allard et al., 1999*; *Doyon et al., 2004*; *Wang et al., 2018a*; *Xu et al., 2016*). The DOM complex (DOM.C) appeared then to be a chimera, a fusion between two complexes with different biochemical activities. It has been proposed that both enzymatic activities of DOM.C, histone acetylation and histone variant exchange, are required for H2A.V turnover during DNA damage response (*Kusch et al., 2004*). It is unclear, however, if this model of DOM.C action could be generalized to other processes, such as transcription regulation. Furthermore, it is still not known how H2A.V is incorporated globally into chromosomes while, at the same time, enriched at promoters.

It is long known that the *dom* transcripts are alternatively spliced to generate two major isoforms, DOM-A and DOM-B (*Ruhf et al., 2001*). We and others previously found that the two splice variants play non-redundant, essential roles during development with interesting phenotypic differences (*Börner and Becker, 2016*; *Liu et al., 2019*).

In this work, we systematically characterized the molecular context and function of each DOM splice variant in *D. melanogaster* cell lines and assessed their contribution to the activity of the DOM.C in the context of transcription. We discovered the existence of two separate, isoform-specific complexes with characteristic composition. Both are involved in transcription regulation, but through different mechanisms. On the one hand, we found that the DOM-B.C is the main ATP-dependent remodeler for H2A.V, responsible for its deposition across the genome and specifically at active promoters. On the other hand, we discovered that DOM-A.C is not involved in bulk H2A.V incorporation, despite the presence of an ATPase domain and many shared subunits with DOM-B.C. Rather, we realized that DOM-A.C might be the 'missing' acetyltransferase NuA4.C of *D. melanogaster*, which specifically targets lysine 12 of histone H4 (H4K12), the most abundant and yet

uncharacterized H4 acetylation in flies (*Feller et al., 2015*). Surprisingly, our data also suggest that the ATPase activity of DOM-A is dispensable for H4K12 acetylation by the DOM-A.C, a principle that might be conserved across metazoans. Our work illustrates how alternative splicing generates functional diversity amongst chromatin regulators.

## Results

### The splice variants of DOMINO, DOM-A and DOM-B, define two distinct complexes

The isoforms of the DOMINO ATPase, DOM-A and DOM-B, are identical for the first 2008 amino acids, but alternative splicing diversifies their C-termini (*Figure 1A*). Both proteins share an N-terminal HSA domain and a central, INO80-like ATPase domain. DOM-A has a longer C-terminus characterized by a SANT domain and a region rich in poly-glutamine stretches (Q-rich). The shorter C-terminus of DOM-B, instead, folds in no predictable manner. Given these differences, we wondered if the interaction partners of the two isoforms might differ. To avoid artefactual association of DOM isoforms with proteins upon overexpression, we inserted a 3XFLAG tag within the endogenous *dom* gene in *D. melanogaster* embryonic cell lines using CRISPR/Cas9. The sites were chosen such that either DOM-A (DOM-RA) or DOM-B (DOM-RE) would be tagged at their C-termini. Of note, the editing of DOM-A C-terminus results in the additional tagging of a longer, DOM-A-like isoform (DOM-RG, which compared to DOM-RA has an insertion of 35 residues at its N-terminus starting from residue 401), but leaves a second DOM-A-like isoform untagged (DOM-RD, 16 residues shorter than DOM-RA at the very C-terminus). We obtained three different clonal cell lines for each isoform (3 homozygous clones for DOM-A, 2 homozygous and 1 heterozygous clone for DOM-B) (*Figure 1—figure supplement 1A,B*). The *dom* gene editing resulted in the expression of 3XFLAG-tagged proteins of the correct size and with similar expression levels across clones (*Figure 1B*).

To identify the strongest and most stable interactors, we enriched the isoforms and associated proteins from nuclear extracts by FLAG-affinity chromatography under very stringent conditions. Mass-spectrometry analysis revealed 13 and 12 strongly enriched interactors (FDR < 0.05 and log2 fold-change >0) for DOM-A and DOM-B, respectively (*Figure 1C*, *Supplementary file 1*). Of those, 7 are common between the two isoforms and were previously characterized as DOM interactors (*Kusch et al., 2004*). Two of the expected subunits, PONT and REPT, associated more strongly with DOM-B than with DOM-A under these conditions (log2 DOM-A IP/CTRL = 1.19 and 1.16, FDR = 0.373 and 0.338). A newly identified DOM-B interactor, HCF, also interacts less strongly DOM-A (log2 DOM-A IP/CTRL = 0.70, FDR = 0.466). The unique interactors revealed interesting differences between DOM-A and DOM-B (*Figure 1D*). Three of the proteins that specifically associate with DOM-A [ING3, E(Pc) and TIP60] share extensive homology with the acetyltransferase module of the yeast NuA4 complex. Another component of the yeast NuA4.C, NIPPED-A, specifically associates with DOM-A. Additionally, we found two transcription factors, XBP1 and CG12054, amongst the DOM-A specific interactors. On the DOM-B side, only ARP6 and PPS appear to be specific interactors of this isoform. While ARP6 was not described before in *Drosophila,* its yeast homolog is essential for H2A.Z remodeling by the SWR1.C (*Wu et al., 2005*). To validate the DOM-A/TIP60 interaction, we raised monoclonal antibodies against TIP60. Co-immunoprecipitation confirmed that TIP60 interacts with DOM-A and not with DOM-B (*Figure 1E* and *Figure 1—figure supplement 1C*). The immunoprecipitation of DOM-A appears to be more efficient when probing with the anti-FLAG antibody compared to the anti-DOM-A polyclonal antibody. This difference might be explained by the presence of one of the DOM-A-like isoforms (DOM-RD), which was left untagged. This isoform is therefore not immunoprecipitated by the anti-FLAG antibody, but it is recognized by the DOM-A specific antibody. Importantly, the same co-immunoprecipitation showed that DOM-A and DOM-B do not interact with each other under these conditions. Taken together, these findings document the existence of two distinct DOM complexes: DOM-A.C and DOM-B.C.

### Specific effects of DOM isoforms on transcription

Previous observation in flies suggested that DOM-A and DOM-B have different, non-redundant functions during *Drosophila* development (*Börner and Becker, 2016*; *Ruhf et al., 2001*). Isoform-specific depletion by RNA interference (RNAi) of either DOM variant in a *Drosophila* embryonic cell line did

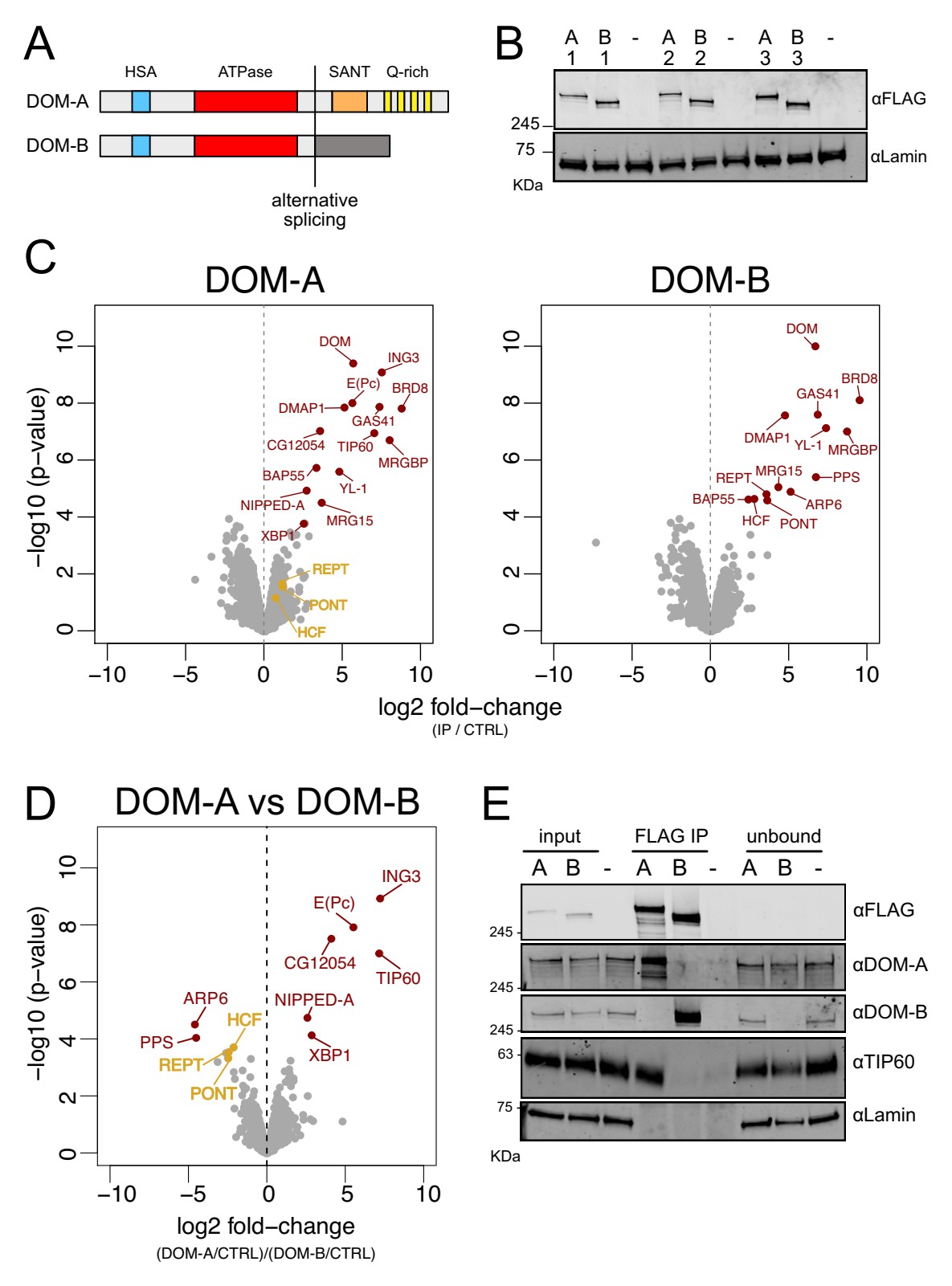

**Figure 1.** DOMINO isoform-specific affinity enrichment reveals distinct DOM-A and DOM-B complexes. (**A**) Schematic representation of the DOM-A (*dom-RA*) and DOM-B (*dom-RE*) isoforms. The two proteins are derived through alternative splicing and differ in their C-termini. (**B**) Western blot showing the expression of 3XFLAG-tagged DOM-A and DOM-B in nuclear fractions derived from three different clonal S2 cell lines (A = DOM A, B = DOM B). Endogenously tagged proteins were detected using αFLAG antibodies. Nuclear extract from S2 cells lacking the tag (-) serves as negative

*Figure 1 continued on next page*

*Figure 1 continued*

control. Lamin: loading control. (**C**) Volcano plot showing -log10 p-value in relation to average log2 fold-change (n = 3 biological replicates) comparing FLAG AP-MS from 3XFLAG DOM-A or DOM-B cell lines (IP) versus 'mock' purifications from untagged S2 cells (CTRL). Red dots represent enriched proteins with FDR < 0.05 and log2 fold-change >0. Orange dots represent proteins significantly enriched in DOM-B AP-MS but not considered as DOM-B specific interactors. (**D**) Volcano plot as described in (C) comparing DOM-A FLAG AP-MS (DOM-A/CTRL) and DOM-B FLAG AP-MS (DOM-B/CTRL). Positive log2 fold-change indicate enrichment in DOM-A pulldown, while negative log2 fold-change indicate enrichment in DOM-B pulldown. Red dots represent isoform-specific enriched proteins with FDR < 0.05. Orange dots represent proteins enriched in DOM-B AP-MS but not considered as DOM-B specific interactors, due to lower statistical significance (FDR > 0.05). (**E**) Western blot validating the mass-spectrometry results. DOM-A clone #2 and DOM-B clone #1 cells were used. Untagged cells (-) serve as negative control. 2% of input and unbound fractions was loaded. Proteins were detected using the antibodies described in the panel.

The online version of this article includes the following figure supplement(s) for figure 1:

**Figure supplement 1.** DOMINO isoform-specific affinity enrichment reveals distinct DOM-A and DOM-B complexes.

not lead to depletion of the other isoform (*Figure 2A*). Interestingly, knock-down of DOM-A (but not DOM-B) led to a strong reduction of TIP60 protein levels. *tip60* mRNA levels were unchanged (*Figure 2—figure supplement 1A*), indicating that TIP60 requires DOM-A for stability (*Figure 2A*). This suggests that most of TIP60 resides in the DOM-A complex in *Drosophila* cells.

The yeast SWR-1 and NuA4 complexes are both implicated in transcription (*Morillo-Huesca et al., 2010*; *Searle et al., 2017*). We therefore explored the functional differences of the two DOM isoforms on transcription by RNAseq. In our analysis, we also included knock-downs of H2A.V and TIP60. Knock-down of either DOM-A, DOM-B, TIP60 or H2A.V individually resulted in significant perturbation of transcription, with notable differences (*Figure 2—figure supplement 1B*, *Supplementary file 2*). Principal Components Analysis (PCA) revealed clearly different transcriptional responses upon loss of DOM-A or DOM-B (*Figure 2B*), which can be visualized by comparing their log2 fold-changes relative to control (*Figure 2C*). The correlation value of 0.45 indicates that many genes are regulated similarly by both ATPases, but a significant number of genes are also differentially affected upon specific depletion of either DOM-A or DOM-B (*Figure 2—figure supplement 1B*). As expected, the transcriptional effects of DOM-A knock-down, but not of DOM-B, resemble the ones caused by knock-down of TIP60 (*Figure 2B,D*). Depletion of H2A.V led to a global reduction of transcription, only observable by normalization to spiked-in *D.virilis* RNA (see methods) (*Figure 2—figure supplement 1C*). The effects of H2A.V depletion were better correlated to DOM-B (r = 0.51) than to DOM-A knock-down (r = 0.25) (*Figure 2B,D*). Many effects of DOM-B depletion may be explained by its H2A.V deposition function, but the ATPase also affects transcription through different routes.

In summary, we found that the depletion of the two DOM isoforms in cells caused specific transcriptional perturbations. The partially overlapping responses upon DOM-B and H2A.V depletions motivated a more in-depth analysis of the relationship between DOM-B and H2A.V levels in the genome.

## The DOM-B complex is the main ATP-dependent remodeler for H2A.V

Both SWR1-type ATPases, DOM-A and DOM-B may contribute to H2A.V incorporation and turnover. We explored global changes in H2A.V levels upon isoform-specific RNAi in nuclear extracts containing chromatin and soluble nuclear proteins. We found a strong H2A.V reduction upon DOM-B depletion (*Figure 3A*, *Figure 3—figure supplement 1A*), while H2A.V mRNA level was unchanged (*Figure 3—figure supplement 1B*). Among the interactors found in our mass-spectrometry analysis, only RNAi against the DOM-B.C-specific subunit ARP6 reduced H2A.V levels to a similar extent (*Figure 3—figure supplement 1C,D*). H2A.V was not affected by the knock-down of DOM-A, TIP60 or other DOM-A.C-specific subunits (*Figure 3A*, *Figure 3—figure supplement 1A,C,D*).

While western blots reveal global changes, they are not sufficiently sensitive to detect changes of H2A.V occupancy at specific sites in chromatin. We therefore employed a more sensitive chromatin immunoprecipitation (ChIP-seq) approach, in which we included *D. virilis* spike-in cells to quantify global changes in H2A.V levels. As expected, we scored dramatic effects on H2A.V levels along the entire genome upon depletion of DOM-B, including promoters and transcriptional termination sites (*Figure 3B,C*, *Figure 3—figure supplement 1E*). Depletion of DOM-A did not affect chromosomal

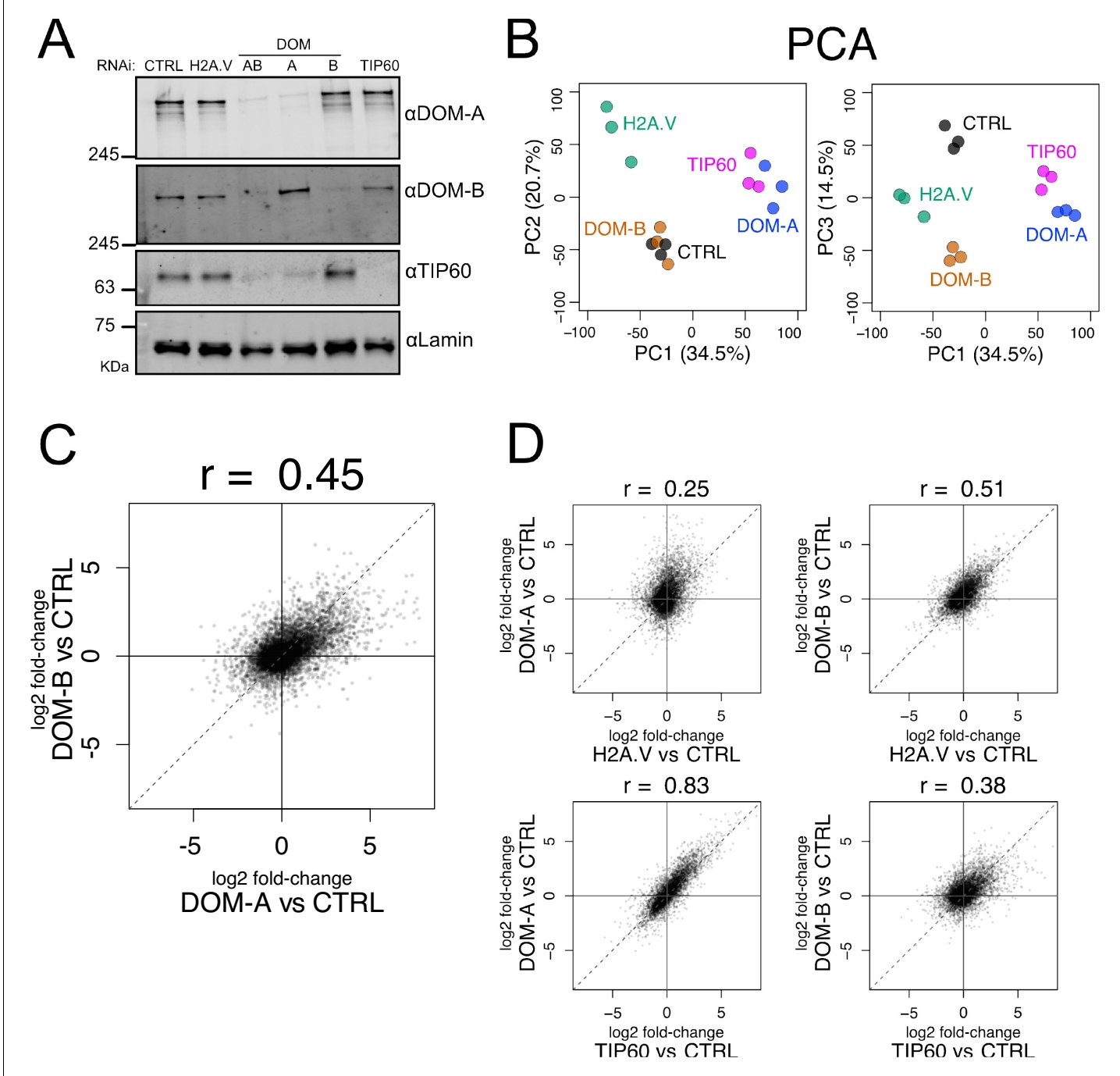

**Figure 2.** Isoform-specific depletion of DOM-A and DOM-B causes distinct transcriptional effects. (**A**) Western blot showing the expression of DOM-A, DOM-B and TIP60 in nuclear extracts of Kc167 cells treated with dsRNA against GST (CTRL), H2A.V, both DOM isoforms (AB), DOM-A (A), DOM-B (B) and TIP60. Proteins were detected with specific antibodies. Lamin: loading control. (**B**) Principal Component Analysis (PCA) comparing transcriptome profiles derived from Kc167 cells treated with dsRNA against GST or GFP (CTRL), H2A.V, DOM-A, DOM-B and TIP60 (n = 3 biological replicates). Three components (PC1, PC2 and PC3) are shown. Percentage of variance is indicated in parenthesis. (**C**) Scatter plot comparing log2 fold-changes in expression of DOM-A against CTRL RNAi and log2 fold-changes in expression of DOM-B against CTRL RNAi for every gene analyzed (N = 10250). Spearman's correlation coefficient (r) is shown above the plot. (**D**) Same as (**C**) but depicting the comparison between DOM-A or DOM-B RNAi and H2A.V or TIP60 RNAi.

The online version of this article includes the following figure supplement(s) for figure 2:

**Figure supplement 1.** Isoform-specific depletion of DOM-A and DOM-B causes distinct transcriptional effects.

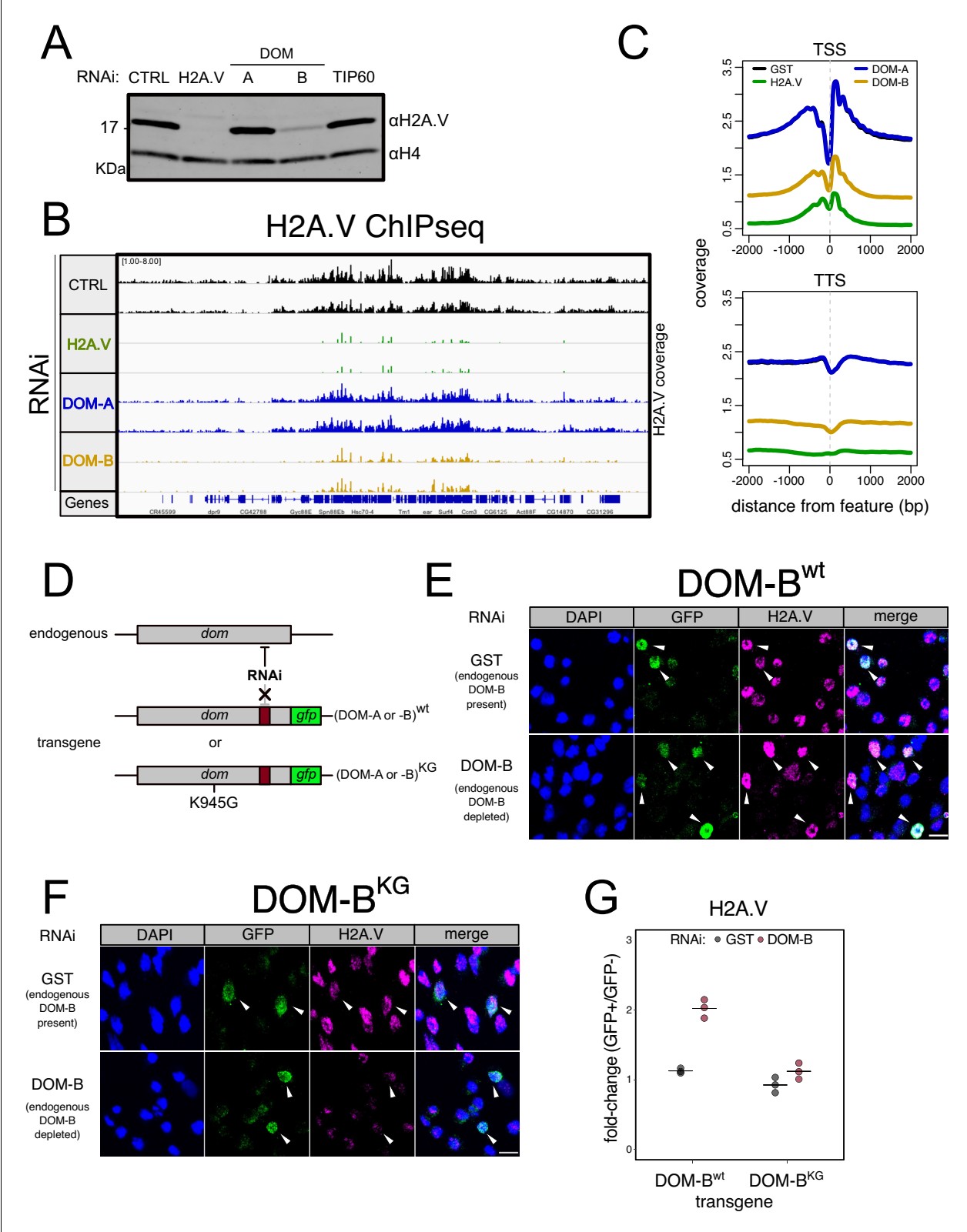

**Figure 3.** DOM-B is responsible for H2A.V incorporation into chromatin in an ATP-dependent manner. (**A**) Western blot showing the expression of H2A. V in nuclear extracts derived from Kc167 cells treated with dsRNA against GST (CTRL), H2A.V, DOM-A (A), DOM-B (B) and TIP60. Histone H4 (H4): loading control. (**B**) Screenshot of genome browser illustrating a region on Chromosome 3R. Each track shows the spike-in and input normalized H2A.V ChIPseq signal derived from Kc167 cells treated with dsRNA as in (**A**). Two biological replicates are shown. (**C**) Composite plot showing spike-in and

*Figure 3 continued on next page*

*Figure 3 continued*

input normalized H2A.V coverage around Transcription Start Sites (TSS) and Transcription Termination Sites (TSS) (N = 10139). Each represent the average coverage (n = 2 biological replicates) of H2A.V in Kc167 cells treated with dsRNA as described in (**A**). (**D**) Schematic representation of the experimental setup to test the requirement for ATPase activity of DOM (A or B) for functionality. A transgene encoding a GFP-tagged wild type or mutant (K945G) DOM is codon-optimized to be resistant to specific dsRNA targeting. The transgene is transfected into Kc167 cells while the endogenous DOM (A or B) are depleted by RNAi. (**E**) Representative immunofluorescence pictures for the DOM-B complementation assay. Cells were treated either with control (GST) dsRNA (endogenous DOM-B present) or with a dsRNA targeting only the endogenous DOM-B (endogenous DOM-B depleted). Cells were transfected with a wild-type transgene encoding RNAi-resistant DOM-B. Cells were stained with DAPI and with GFP and H2A.V antibodies. Arrows indicate the cells where the transgene is expressed and nuclear. Scale bar: 10 μm **F** Same as (**E**) but cells were transfected with a mutant DOM-B (K945G) (**G**) Dot plot showing the quantification of the immunofluorescence-based complementation assay. Each dot represents the fold-change of mean H2A.V signal between GFP-positive (in which the transgene is expressed) and GFP-negative cells in one biological replicate (>100 total cells/replicate). Cells were treated with dsRNAs as in (**E**) Wild-type or mutant DOM-B transgenes are compared.

The online version of this article includes the following figure supplement(s) for figure 3:

**Figure supplement 1.** DOM-B is responsible for H2A.V incorporation into chromatin in an ATP-dependent manner.

H2A.V at any of these sites (*Figure 3B,C*). These observations support the notion that the DOM-B.C, and not the DOM-A.C, is the remodeler dedicated to H2A.V incorporation.

Since SWR1-type remodelers bind and hydrolyze ATP to incorporate H2A.Z variants (*Hong et al., 2014*; *Willhoft et al., 2018*), we wanted to confirm the ATP-requirement for in vivo incorporation of H2A.V by DOM-B. We devised an RNAi-based complementation strategy in which we rescued the effects of depleting endogenous *dom-B* mRNA by expression of RNAi-resistant *dom-B* transgenes. The functional complementation involved wild-type DOM-B or a mutant predicted to be deficient in ATP-binding (K945G) (*Hong et al., 2014*; *Mizuguchi et al., 2004*; *Figure 3D*). GFP-tagging of the DOM-B proteins allowed to selectively monitor the H2A.V levels by immunofluorescence microscopy in the cells in which the transgenes are expressed. We detected higher levels of H2A.V in cells complemented with a wild-type DOM-B transgene, indicating the expected rescue (*Figure 3E*). Conversely, the remodeling-defective mutant transgene did not increase the residual H2A.V levels (*Figure 3F*). Comparing the mean H2A.V signal between cells that express the transgene (GFP+) and cells that don't (GFP-), revealed once more that only the wild-type could restore H2A.V levels (*Figure 3G*, *Figure 3—figure supplement 1F*). The data suggest that the DOM-B.C is responsible for the incorporation of H2A.V in an ATP-dependent manner.

## The DOM-A complex is related to the yeast NuA4 complex and catalyzes H4K12 acetylation

Despite the presence of an ATPase domain identical to DOM-B, DOM-A does not seem to be responsible for H2A.V incorporation in steady state. Therefore, we considered other functions for DOM-A.C. The striking correlation between transcriptional responses upon TIP60 and DOM-A depletion suggests a unique association with functional relevance. Our mass-spectrometry analysis had identified several proteins that are homologous to corresponding subunits of the yeast NuA4 HAT complex. The core NuA4.C subunit EAF1 is a small protein with prominent N-terminal HSA and C-terminal SANT domains. DOM-A also features similarly arranged domains, but they are separated by the long ATPase domain (*Figure 4—figure supplement 1A*). This raises the question whether DOM-A might serve as the central subunit of a NuA4-type complex in *Drosophila*. The existence of such a complex with functional and structural similarity to the well-studied yeast complex has not been reported so far. Since NuA4.C is responsible for histone acetylation, we looked at H3 and H4 acetylation changes upon DOM isoform-specific knock-down by targeted mass-spectrometry.

The hypothesis of a *Drosophila* NuA4.C poses the acetyltransferase TIP60 as the main effector of DOM-A.C. This is supported by the earlier finding that TIP60 is unstable in the absence of DOM-A (*Figure 2A*). We therefore included TIP60 knock-down for our targeted mass-spectrometry. Our analysis showed that RNAi against DOM-A, but not against DOM-B, specifically reduces H4K12ac (average 28.9% reduction) and, to a lesser extent, H4K5ac (average 23.1% reduction) (*Figure 4A*, *Figure 4—figure supplement 1B*). Importantly, unsupervised clustering shows that depleting DOM-A or TIP60 lead to very similar changes in histone acetylation patterns: depletion of TIP60 also reduces H4K12ac, by on average 36.3% and H4K5ac by 16.4% (*Figure 4A*, *Figure 4—figure supplement 1B*). Interestingly, we detected a decrease in monomethylation and increase of trimethylation

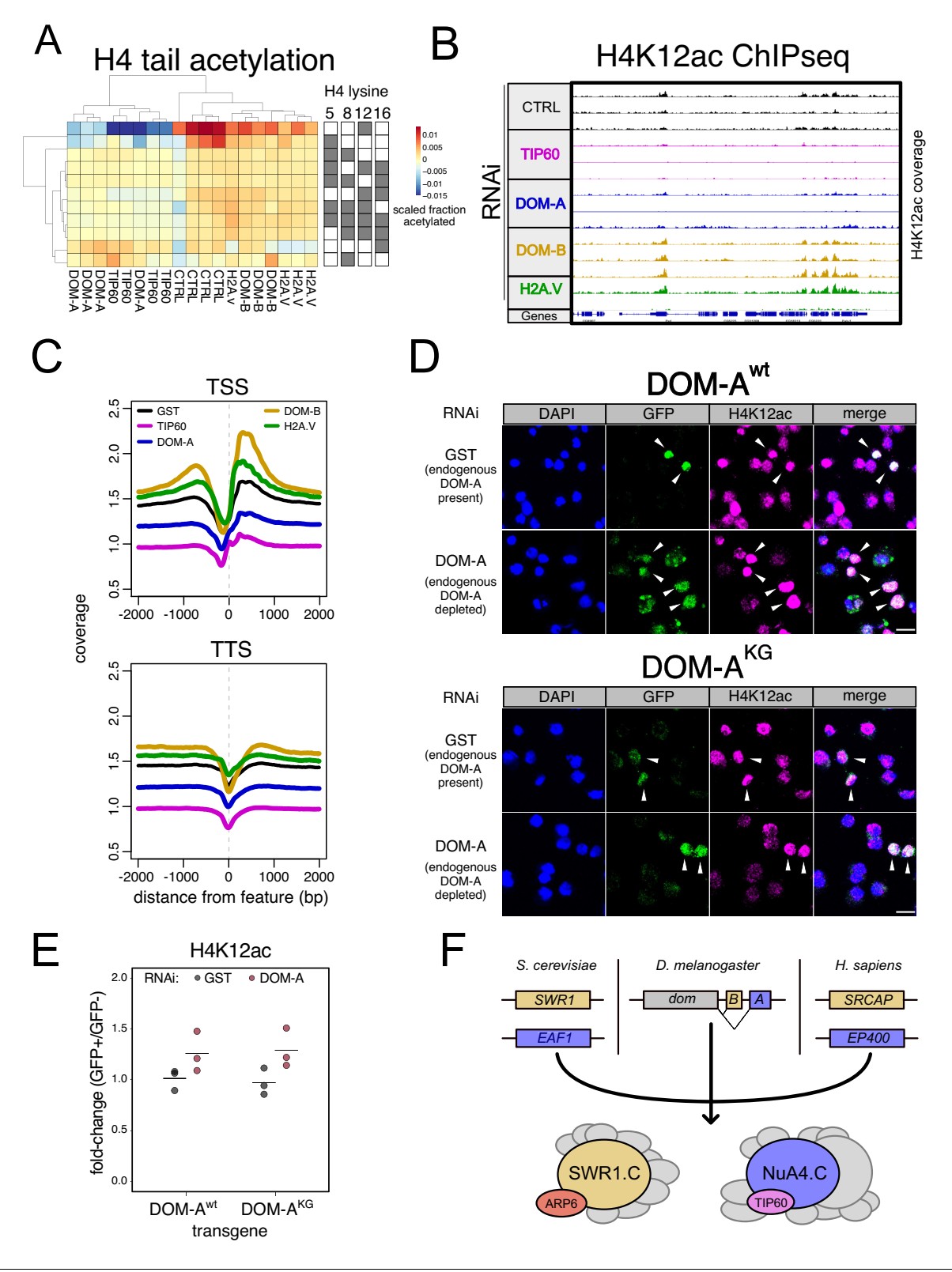

**Figure 4.** The DOM-A.C acetylates H4K12. (**A**) Heatmap shows scaled acetylation levels for various histone H4 residues (measured by mass-spectrometry) in Kc167 cells treated with dsRNA against GST or GFP (CTRL), H2A.V, DOM-A (A), DOM-B (B) and TIP60. Individual biological replicates are shown. Rows and columns are clustered based on Euclidean distance. (**B**) Screenshot of genome browser illustrating a region on Chromosome 3R. Each track shows spike-in and input normalized H4K12ac ChIPseq signal derived from Kc167 cells treated with dsRNA against GST or GFP (CTRL),

*Figure 4 continued on next page*

*Figure 4 continued*

TIP60, DOM-A, DOM-B and H2A.V. 3 biological replicates are shown for all RNAi except H2A.V (2 biological replicates). (**C**) Composite plot showing spike-in and input normalized H4K12ac coverage around Transcription Start Sites (TSS) and Transcription Termination Sites (TSS) (N = 10139). Each represent the average coverage (n = 2 biological replicates for H2A.V RNAi, n = 3 biological replicates for all the other knock-down) of H4K12ac in Kc167 cells treated with dsRNA as described in (**A**). (**D**) Representative immunofluorescence pictures for the DOM-A complementation assay. Cells were treated either with control (GST) dsRNA (endogenous DOM-A present) or with a dsRNA targeting the endogenous DOM-A (endogenous DOM-A depleted). In the top panel, cells were transfected with a wild-type transgene encoding for DOM-A. In the bottom panel, cells were transfected with a mutant (K945G) DOM-A transgene. Cells were stained with DAPI, and with GFP and H4K12ac antibodies. Arrows indicate the cells where the transgene is expressed and nuclear. Scale bar: 10 µm (**E**) Dot plot showing the quantification of the immunofluorescence-based complementation assay. Each dot represents the fold-change of mean H4K12ac signal between GFP-positive (in which the transgene is expressed) and GFP-negative cells in one biological replicate (>100 total cells/replicate). Cells were treated with dsRNAs as in (**E**) Wild-type or mutant DOM-A transgenes are compared. (**F**) Model for SWR1.C and NuA4.C specification in *S. cerevisiae*, *D. melanogaster* and *H. sapiens*.

The online version of this article includes the following figure supplement(s) for figure 4:

**Figure supplement 1.** The DOM-A.C acetylates H4K12.

at H3K27 by DOM depletion, a bit stronger for DOM-A or TIP60 (*Figure 4—figure supplement 1C, D*). H4 methylation was unchanged (*Figure 4—figure supplement 1E*).

The H4K12 seems to be the most prominent chromatin target of the DOM-A/TIP60 complex. We sought to confirm the mass spectrometric result by an orthogonal ChIP-seq experiment. We found the H4K12ac signal reduced in many regions of the genome, including promoters and transcriptional termination sites, upon TIP60 RNAi and to a lesser extent upon DOM-A RNAi (*Figure 4B,C*, *Figure 4—figure supplement 1F,H,I*), but the results suffer from variability, probably due to a low ChIP efficiency of the H4K12ac antibody (*Figure 4—figure supplement 1F*). Comparison between H4K12ac and transcription showed that genes downregulated in DOM-A or TIP60 knock-down tend to have higher basal levels of H4K12ac at promoters (*Figure 4—figure supplement 1G*). The reduction of H4K12ac by DOM-A or TIP60 knock-down, however, is global and does not affect some genes specifically (*Figure 4—figure supplement 1H*). Remarkably, depletion of DOM-B caused an unexpected increase in H4K12 acetylation at many chromosomal regions that lose the mark upon DOM-A ablation (*Figure 4B,C*). Depletion of H2A.V also causes a small H4K12ac increase similarly to DOM-B knock-down, as if the absence of this remodeler and/or its substrate allowed more DOM-A activity at promoters.

The yeast NuA4.C does not contain a functional ATPase at its core. To explore whether the acetylation of H4K12 catalyzed by the DOM-A.C depends on ATP-dependent nucleosome remodeling activity, we employed the same RNAi-based complementation strategy we had used for DOM-B (*Figure 3D*). DOM-A was depleted and RNAi-resistant DOM-A wild-type or ATPase mutant derivatives were tested for their ability to rescue the loss of H4K12ac. As expected, the wild-type DOM-A transgene restored H4K12 acetylation (*Figure 4D*). Remarkably, this acetylation did not depend on a functional DOM-A ATPase. Comparison of the mean H4K12ac signals in cells that do or do not express the transgene confirmed that both, the wild-type and the mutant transgenes, could restore H4K12ac levels (*Figure 4E*, *Figure 4—figure supplement 1J*).

## Discussion

Our mass-spectrometry analysis of endogenously expressed DOM isoforms purified under stringent conditions revealed two separate complexes. A DOM.C was previously reported after overexpression and affinity-purification of tagged PONTIN (*Kusch et al., 2004*), which yielded a mixture of DOM-A and DOM-B complexes and may be contaminated with the dINO80 complex, which also contains PONTIN (*Klymenko et al., 2006*). In light of our results, we think the model for H2A.V exchange during DNA damage response proposed in this early work (*Kusch et al., 2004*) should be re-visited accounting for the contribution of both DOM-A.C, DOM-B.C and possibly dINO80.C. It will be interesting to define the role of each complex on the recognition and restoration of damaged chromatin, especially at the level of H2A.V remodeling and acetylation-based signaling.

We previously showed that DOM-B, and not DOM-A or TIP60, affects H2A.V levels during fly oogenesis (*Börner and Becker, 2016*). In addition to confirming this finding in a different system and with complementary experimental approaches, we now also showed that the DOM-B.C is

responsible for H2A.V incorporation into chromatin. The reaction requires ATP, like SWR1.C-mediated H2A.Z incorporation. We also discovered a previously unidentified subunit, ARP6, which is necessary for the maintenance of H2A.V global levels, just like DOM-B. In yeast SWR1.C and human SRCAP.C, the ARP6 orthologs are indispensable for nucleosome remodeling since they couple the ATPase motor to productive nucleosome sliding (*Matsuda et al., 2010*; *Willhoft et al., 2018*; *Willhoft and Wigley, 2020*; *Wu et al., 2005*). The *Drosophila* DOM-B.C is likely to employ a similar remodeling mechanism. Knock-down of DOM-B affects transcription, but the effects overlap only partially with those that follow H2A.V depletion. This discrepancy could be explained in several ways. First, the reduction of H2A.V levels upon DOM-B knock-down is not as extensive as the one caused by direct depletion of H2A.V. The residual levels of H2A.V upon DOM-B depletion may suffice to regulate transcription at many promoters. Second, we cannot exclude that DOM-B.C also impacts transcription independently of H2A.V incorporation. Third, the global increase of H4K12ac at promoters upon DOM-B knock-down might indirectly compensate for the loss of H2A.V at some specific genes.

The DOM-A.C, surprisingly, did not affect H2A.V incorporation under physiological conditions in any of our assays, in agreement with what has been observed for the DOM-A isoform during oogenesis (*Börner and Becker, 2016*). DOM-A.C lacks the ARP6 subunit that is a mechanistic requirement for nucleosome remodeling by INO80-type remodelers (*Willhoft and Wigley, 2020*). Because the ATPase domain of DOM-A is identical to the one in DOM-B, it is possible that DOM-A utilizes ATP under circumstances that we did not monitor in our study. It is also possible that DOM-A.C-specific subunits have an inhibitory effect on DOM-A ATPase activity through allosteric regulation. Of note, the recombinant human ortholog of DOM-A, EP400, can incorporate H2A.Z (*Park et al., 2010*), but H2A.Z levels are unaffected if EP400 is depleted in vivo, where it resides in a multi-subunit complex (*Pradhan et al., 2016*). Regulation of nucleosome remodeling through autoinhibitory domains or associated subunits is a widespread mechanism (*Clapier et al., 2017*).

Our data suggest that the DOM-A.C is the functional equivalent of the yeast NuA4.C, which acetylates the H4 N-terminus (*Kuo et al., 2015*) and possibly other proteins. Depletion of DOM-A.C causes a significant reduction of H4K12ac at a global level. Some genomic regions that still retain a high H4K12ac ChIP signal in the absence of DOM-A may be explained by the presence of additional acetyltransferases targeting H4K12, such as CHAMEAU (*Feller et al., 2015*; *Peleg et al., 2016*). The function of H4K12ac is still largely unknown in *Drosophila*, although it has been implicated in aging (*Peleg et al., 2016*). We speculate that H4K12ac may participate in transcriptional regulation since knock-down of DOM-A or TIP60 perturb the transcriptional program in a very similar manner. Genes down-regulated upon DOM-A and TIP60 RNAi show high H4K12ac around their TSS, but the H4K12ac is not specifically reduced at their promoters. We speculate that these genes might rely more on H4K12ac for their expression or be more sensitive to changes in acetylation. Given that H4K12ac is the most abundant H4 acetylation (*Feller et al., 2015*; *Peleg et al., 2016*), we also cannot exclude that some of these effects are due to global and aberrant chromosomal condensation. The increase in H4K12ac at promoter observed upon DOM-B RNAi appears to be partially phenocopied by the knock-down of H2A.V. It is possible that H2A-containing nucleosomes are a better substrate for the DOM-A.C compared to the ones containing only H2A.V. The loss of the variant might therefore result in higher H4K12ac catalyzed by TIP60.

In the absence of DOM-A, TIP60 is unstable suggesting that the DOM-A.C is the major form of TIP60, at least in *D. melanogaster* cells. It also suggests that during evolution a HAT module and some components of the SWR1.C became stably associated in a new functional complex, the dNuA4.C, as it has been proposed for the human EP400 complex (*Auger et al., 2008*). An intermediate case is found in *C. albicans*, where acetylation of EAF1 by TIP60 mediates a reversible association between the NuA4.C and SWR1.C (*Wang et al., 2018b*). We found that the DOM-A.C H4K12 HAT activity does not need the DOM-A ATPase activity. In yeast, inserting the ATPase domain of the *Drosophila* DOM between the HSA and SANT domains of EAF1, the central subunit of the yeast NuA4, does not affect its function (*Auger et al., 2008*). Such a situation could also apply to DOM-A. Lastly, our mass-spectrometry analysis revealed new, uncharacterized interactors for DOM-A. Of those, the transcription factor CG12054 has been found as a potential DOM partner in a previous screen (*Rhee et al., 2014*). Its human ortholog, JAZF1, appears to be involved in transcriptional repression (*Nakajima et al., 2004*) and has been associated to endometrial stromal tumors (*Koontz et al., 2001*). Its function in flies is unknown.

Division of labor between chromatin modifying enzymes is key to ensure efficient regulation of nuclear processes. During evolution, genome duplications and genetic divergence expand and diversify activities. The case of DOM illustrates beautifully how evolution can take different routes to achieve what must be assumed as an important functional specification (*Figure 4F*, *Supplementary file 3*). In yeast, the SWR1 and NuA4 complexes are entirely separate entities. In humans, whose genomes underwent duplication events, the paralogous SRCAP and EP400 protein ATPases each organize different complexes that may serve distinct, conserved functions. In *Drosophila* a similar specialization was achieved by alternative splicing. Surprisingly, the gene orthologs of *dom* in honeybee (*A. mellifera*, LOC413341), jewel wasp (*N. vitripennis,* LOC100115939), Jerdon's jumpin ant (*H. saltator*, LOC105183375), red flour beetle (*T. castaneum*, LOC656538) and even in the common house spider (*P. tepidariorum*, LOC107448208) undergo alternative splicing to generate at least two isoforms with different C-termini. The mode of specification of SWR1 and NuA4 through splice variants might therefore not be limited to *Drosophila,* but more wide-spread throughout the Arthropoda phylum.

# Materials and methods

## Key resources table

| Reagent type (species) or resource | Designation | Source or reference | Identifiers | Additional information |
| --- | --- | --- | --- | --- |
| Cell line (*D. melanogaster*) | Kc167 | DGRC | FLYB; FBtc0000001 | |
| Cell line (*D. melanogaster*) | S2 (Clone L2-4) | *Villa et al., 2016* | | Gift from P Heun lab |
| Cell line (*D. virilis*) | 79f7Dv3 | *Albig et al., 2019* | | Gift from BV Adrianov |
| Antibody | DOM-A (17F4) (rat monoclonal) | *Börner and Becker, 2016* and this publication | | 1:5 (WB) |
| Antibody | DOM-A (SA-8977) (rabbit polyclonal) | This publication | | 1:1000 (WB) |
| Antibody | DOM-B (SA-8979) (rabbit polyclonal) | This publication | | 1:1000 (WB) |
| Antibody | TIP60 (11B10) (rat monoclonal) | This publication | | 1:20 (WB) |
| Antibody | H2A.V (Rb-H2Av) (rabbit polyclonal) | *Börner and Becker, 2016* | | 1:1000 (WB) 1:2500 (IF) 25 µl/IP (ChIP) |
| Antibody | H4 (rabbit polyclonal) | abcam | ab10158 | 1:5000 (WB) |
| Antibody | H4K12ac (rabbit polyclonal) | Merck-Millipore | 07–595 | 1:2500 (IF) 2 µl/IP (ChIP) |
| Antibody | FLAG-m2 (mouse monoclonal) | Sigma-Aldrich | F3165 | 1:1000 (WB) |
| Antibody | GFP (mouse monoclonal) | Roche | 11814460001 | 1:500 (IF) |
| Antibody | Lamin (mouse monoclonal) | Gift from H Saumweber | | 1:1000 (WB) |
| Recombinant DNA reagent | pENTR3C (plasmid) | Thermo Fischer Scientific | A10464 | |

*Continued on next page*

*Continued*

| Reagent type (species) or resource | Designation | Source or reference | Identifiers | Additional information |
|---|---|---|---|---|
| Recombinant DNA reagent | pHWG (plasmid) | DGRC | | Kind gift from P Korber |

Plasmids, primers, cell lines and antibodies from this study are available upon request from Peter B Becker (pbecker@bmc.med.lmu.de).

## Cell lines and culture conditions

*D. melanogaster* embryonic Kc167 cell line was obtained from the *Drosophila* Genomic Resource Center (https://dgrc.bio.indiana.edu/Home). *D. melanogaster* S2 (subclone L2-4) cell line was a kind gift of P Heun (*Villa et al., 2016*). *D. virilis* 79f7Dv3 cell line was a kind gift of BV Adrianov (*Albig et al., 2019*). The identity of cell lines was verified by high-throughput sequencing. Cells were subjected to mycoplasma testing. Cells were maintained at 26°C in Schneider's *Drosophila* Medium (Thermo-Fischer, Cat. No 21720024) supplemented with 10% FBS (Kc167 and S2) or 5% FBS (79f7Dv3) and 1% Penicillin-Streptomycin solution (Sigma-Aldrich, Cat No P-4333).

## CRISPR/Cas9 tagging

gRNAs targeting exon 14 (DOM-A and DOM-G, Flybase transcripts *dom-RA* and *dom-RG*) or exon 11 (DOM-B, Flybase transcript *dom-RE*) were initially designed using GPP sgRNA designer (https://portals.broadinstitute.org/gpp/public/analysis-tools/sgrna-design; *Doench et al., 2016*). gRNA candidates were checked for off-targets using flyCRISPR Target Finder (https://flycrispr.org/target-finder/, guide length = 20, Stringency = high, PAM = NGG only) (*Gratz et al., 2014*). Two gRNAs each for DOM-A and DOM-B were selected (*Supplementary file 4*). The 20 bp gRNAs were fused to a tracrRNA backbone during synthesis and cloned downstream of the *Drosophila* U6 promoter. These constructs were synthesized as gBlocks (Integrated DNA Technologies) and PCR-amplified before transfection using Q5 polymerase (New England Biolabs, Cat No. M0491S). To generate the repair template, the sequence encoding for 3XFLAG tag, including a stop codon, was inserted between two homology arms of 200 bp each by gene synthesis (Integrated DNA Technologies) (*Supplementary file 4*). The repair templates were cloned in pUC19 to generate repair plasmids. For CRISPR editing, one million S2 cells (subclone L2-4) in 500 µL medium were seeded in each well of a 24-well plate. After 4 hr, cells were transfected with 110 ng of gRNAs (55 ng each), 200 ng of repair plasmid and 190 ng of pIB_Cas9_Blast (encoding SpCas9 and carrying Blasticidin resistance, kind gift of P Heun) using X-tremeGENE HP DNA Transfection Reagent (Roche, Cat. No 6366236001). 24 hr after transfection, medium was replaced with 500 µL of fresh medium containing 25 µg/ml Blasticidin (Gibco, Cat. No A1113903). Three days after selection the cells were collected and seeded into 6 cm tissue culture dishes at three different concentrations (1000, 2000 and 5000 cells/well) and allowed to attach for 1–2 hr. Medium was then removed and cells carefully overlaid with 2.5 mL of a 1:1 mix of 2X Schneider's Medium (prepared from powder, Serva, Cat. No 35500) + 20% FBS + 2% Penicillin-Streptomycin and 0.4% low-melting agarose equilibrated to 37°C. The dishes were sealed with parafilm, inserted into 15 cm dishes together with a piece of damp paper and sealed once more with parafilm. After 2 to 3 weeks, individual cell colonies were picked using a pipet, suspended in 100 µL of Schneider's *Drosophila* Medium + 10% FBS + 1% Penicillin-Strepto-mycin and plated into 96-well plates. Clones were expanded for 1–2 weeks and further expanded into 48-well plates. For PCR-testing of clones, 50 µL of cells were collected, 50 µL water was added and DNA was purified using Nucleospin Gel and PCR Cleanup (Macherey-Nagel, Cat. No 740609.250). Extracted DNA was PCR-amplified to check the insertion of the 3XFLAG tag. The PCR product of DOM-A clone #2 results larger due to the presence of an insertion 29 bp downstream of the stop codon (*Figure 1—figure supplement 1B*).

Selected clones were further expanded and stored in liquid nitrogen in 90% FBS + 10% DMSO.

## Nuclear extraction and FLAG affinity enrichment

For nuclear extraction from 3XFLAG-tagged cells lines, 0.5–1 billion cells were collected by centrifugation at 500 g for 5 min. Cells were washed with 10 ml of PBS, resuspended in 10 ml of ice-cold NBT-10 buffer [15 mM HEPES pH 7.5, 15 mM NaCl, 60 mM KCl, 0.5 mM EGTA pH 8, 10% Sucrose, 0.15% Triton-X-100, 1 mM PMSF, 0.1 mM DTT, 1X cOmplete EDTA-free Protease Inhibitor (Roche, Cat. No 5056489001)] and rotated for 10 min at 4°C. Lysed cells were gently overlaid on 20 ml of ice-cold NB-1.2 buffer (15 mM HEPES pH 7.5, 15 mM NaCl, 60 mM KCl, 0.5 mM EGTA pH 8, 1.2 M Sucrose, 1 mM PMSF, 0.1 mM DTT, 1X cOmplete EDTA-free Protease Inhibitor) and spun at 4000 g for 15 min. Pelleted nuclei were washed once with 10 ml of ice-cold NB-10 buffer (15 mM HEPES pH 7.5, 15 mM NaCl, 60 mM KCl, 0.5 mM EGTA pH 8, 10% Sucrose, 1 mM PMSF, 0.1 mM DTT, 1X cOmplete EDTA-free Protease Inhibitor) and resuspended in ice-cold Protein-RIPA buffer (50 mM Tris-HCl pH 7.5, 150 mM NaCl, 1 mM EDTA, 0.5% IGEPAL CA-630, 0.5% Na-Deoxycholate, 0.1% SDS, 1 mM PMSF, 0.1 mM DTT, 1X cOmplete EDTA-free Protease Inhibitor). Nuclei were sonicated in 15 mL Falcons tubes using Diagenode Bioruptor (20 cycles, 30 s ON/30 s OFF). Extract was spun at 16000 g for 15 min at 4°C in a table-top centrifuge. Soluble extract was collected and total protein concentration determined using Protein Assay Dye Reagent Concentrate (BIO-RAD, Cat No 5000006) with BSA as standard. 2 mg aliquots were flash-frozen. For FLAG-immunoprecipitation, 2 mg of nuclear protein were thawed, spun at 160,000 g for 15 min at 4°C to remove aggregates. Extracts were diluted 1:1 with Benzonase dilution buffer (50 mM Tris-HCl pH 7.5, 150 mM NaCl, 4 mM MgCl$_2$, 0.5% NP-40, 1 mM PMSF, 0.1 mM DTT, 1X cOmplete EDTA-free Protease Inhibitor). 60 µL (50% slurry) of Protein-RIPA-equilibrated FLAG-m2 beads (Sigma-Aldrich, Cat. No A2220) were added together with 1 µL of Benzonase (Merck-Millipore, Cat. No 1.01654.0001). After 3 hr of incubation at 4°C on a rotating wheel, the beads were washed 3 times with ice-cold Protein-RIPA buffer and thrice with ice-cold TBS (5 mM Tris-HCl pH 7.5, 150 mM NaCl) (5 min rotation each, 4°C). For western blots, beads were then resuspended in 50 µL of 5X Laemmli Sample buffer (250 mM Tris-HCl pH 6.8, 10% w/v SDS, 50% v/v glycerol, 0.1% w/v bromophenol blue, 10% β-mercaptoethanol) and boiled for 5 min at 95°C. For mass-spectrometry, beads were incubated with 50 µL of elution buffer (2 M urea, 50 mM Tris-HCl, pH 7.5, 2 mM DTT and 10 µg ml$^{-1}$ trypsin) for 30 min at 37°C. The eluate was removed and beads were incubated in 50 µL of alkylation buffer (2 M urea, 50 mM Tris-HCl, pH 7.5 and 10 mM chloroacetamide) at 37°C for 5 min. Combined eluates were further incubated overnight at room temperature. Tryptic-peptide mixtures were acidified with 1% Trifluoroacetic acid (TFA) and desalted with Stage Tips containing three layers of SDB-RPS (Polystyrene-divinylbenzene copolymer partially modified with sulfonic acid) material. To this end, samples were mixed 1:1 with 1% TFA in isopropanol and loaded onto the stagetip. After two washes with 100 µL 1%TFA in Isopropanol and two washes with 100 µl 0.2%TFA in water, samples were eluted with 80 µl of 2% (v/v) ammonium hydroxide, 80% (v/v) acetonitrile (ACN) and dried on a centrifugal evaporator. Samples were dissolved in 10 µL Buffer A* (2% ACN/0.1% TFA) for mass spectrometry. Peptides were separated on 50-cm columns packed in house with ReproSil-Pur C18-AQ 1.9 µm resin (Dr Maisch). Liquid chromatography was performed on an EASY-nLC 1200 ultra-high-pressure system coupled through a nanoelectrospray source to a Q-Exactive HF-X mass spectrometer (Thermo Fisher). Peptides were loaded in buffer A (0.1% formic acid) and separated by application of a non-linear gradient of 5–30% buffer B (0.1% formic acid, 80% ACN) at a flow rate of 300 nl min$^{-1}$ over 70 min. Data acquisition switched between a full scan and 10 data-dependent MS/MS scans. Full scans were acquired with target values of $3 \times 10^6$ charges in the 300–1,650 $m/z$ range. The resolution for full-scan MS spectra was set to 60,000 with a maximum injection time of 20 ms. The 10 most abundant ions were sequentially isolated with an ion target value of $1 \times 10^5$ and an isolation window of 1.4 $m/z$. Fragmentation of precursor ions was performed by higher energy C-trap dissociation with a normalized collision energy of 27 eV. Resolution for HCD spectra was set to 15,000 with a maximum ion-injection time of 60 ms. Multiple sequencing of peptides was minimized by excluding the selected peptide candidates for 30 s. In total, 3 technical replicates (parallel immunoprecipitations) for each of the 3 biological replicates (1 clone = 1 replicate, extract prepared on different days) were analyzed. Raw mass spectrometry data were analyzed with MaxQuant (version 1.5.6.7) (*Cox and Mann, 2008*) and Perseus (version 1.5.4.2) software packages. Peak lists were searched against the *Drosophila melanogaster* UniProt FASTA database combined with 262 common contaminants by the integrated Andromeda search engine (*Cox et al., 2011*). The false

discovery rate (FDR) was set to 1% for both peptides (minimum length of 7 amino acids) and proteins. 'Match between runs' (MBR) with a maximum time difference of 0.7 min was enabled. Relative protein amounts were determined with the MaxLFQ algorithm (*Cox et al., 2014*), with a minimum ratio count of two. Missing values were imputed from a normal distribution, by applying a width of 0.2 and a downshift of 1.8 standard deviations. Imputed LFQ values of the technical replicates for each biological replicate were averaged. Differential enrichment analysis was performed in R using the *limma* package as previously described (*Kammers et al., 2015*; *Ritchie et al., 2015*). Adjusted p-values (FDR) were calculated using the *p.adjust* function (*method = 'fdr'*) (*Source code 1*).

## RNAi

Primers for dsRNA templates were either obtained from the TRiP website (https://fgr.hms.harvard.edu/fly-in-vivo-rnai), designed using SnapDragon (https://www.flyrnai.org/snapdragon) or designed using E-RNAi (https://www.dkfz.de/signaling/e-rnai3/) (*Horn and Boutros, 2010*; *Perkins et al., 2015*), except one primer pair for *Tip60* RNAi which was obtained from *Kusch et al. (2004)* (*Supplementary file 4*). Templates for in vitro transcription were generated by PCR-amplification using Q5 Polymerase (New England Biolabs, Cat No. M0491S). dsRNAs were generated by in vitro transcription using MEGAScript T7 kit (Invitrogen, Cat. No AMB 13345), followed by incubation at 85°C for 5 min and slow cool-down to room temperature. *D. melanogaster* Kc167 cell were collected by spinning at 500 g for 5 min. Cells were washed once with PBS and resuspended in Schneider's *Drosophila* Medium without serum and Penicillin-Streptomycin at a concentration of 1.5 million/ml (for RNAi in 12-well and 6-well plates) or 3 million/ml (for RNAi in T-75 flasks). 0.75 million (12-well), 1.5 million (6-well) or 15 million (T-75 flasks) cells were plated and 5 μg (12-well), 10 ug (6-well) or 50 μg (T-75 flasks) of dsRNA was added. Cells were incubated for 1 hr with gentle rocking and 3 volumes of Schneider's *Drosophila* Medium (supplemented with 10% FBS and 1% Penicillin-Streptomycin solution) was added. After 6 days cells were collected and analyzed.

## RNAseq

Two million of Kc167 cells were pelleted at 500 g for 5 min. Cells were resuspended in 1 mL of PBS and 1 million of *D. virilis* 79f7Dv3 cells were added. Cells were pelleted at 500 g for 5 min and total RNA was extracted using RNeasy Mini Kit (QIAgen, Cat No. 74104), including DNAse digestion step (QIAgen, Cat No. 79254). mRNA was purified using Poly(A) RNA selection kit (Lexogen, Cat. No M039100). Both total RNA and mRNA quality was verified on a 2100 Bioanalyzer (Agilent Technologies, Cat. No G2939BA). Libraries for sequencing were prepared using NEBnext Ultra II directional RNA library prep kit for Illumina (New England Biolabs, Cat. No E7760L). Libraries were sequenced on an Illumina HiSeq 1500 instrument at the Laboratory of Functional Genomic Analysis (LAFUGA, Gene Center Munich, LMU). For the analysis, 50 bp single reads were mapped to the *D. melanogaster* (release 6) or independently to the *D. virilis* (release 1) genome using STAR (version 2.5.3a) with the GTF annotation `dmel-all-r6.17.gtf` or `dvir-all-r1.07.gtf`, respectively. Multi-mapping reads were filtered out by the parameter `–outFilterMultimapNmax 1`. Genic reads were counted with the parameter `–quantMode GeneCounts`. Read count tables were imported to R and low count genes were removed (at least 1 read per gene in 6 of the samples). Normalization factors (`sizeFactors`) were calculated for *D. melanogaster* or *D. virilis* count tables independently using DESeq2 package (version 1.24). Normalization factors derived from *D. virilis* were applied to *D. melanogaster* counts. Statistical analysis was carried out using DESeq2 by providing replicate information as batch covariate. Estimated log2 fold-change and adjusted p-values were obtained by the results function (DESeq2) and adjusted p-value threshold was set 0.01. Batch effect was corrected by the *ComBat* function from the sva package (version 3.32) on the log2-transformed normalized read counts. Batch adjusted counts were plotted relative to control or used for principal component analysis (PCA). Plots were generated using R graphics. Scripts are available on GitHub (https://github.com/tschauer/Domino_RNAseq_2020).

## RT-qPCR

cDNA was synthesized from 1 μg of total RNA (extracted as previously described but omitting the addition of *D. virilis* spike-in cells) using Superscript III First Strand Synthesis System (Invitrogen, Cat. No 18080–051, random hexamer priming) and following standard protocol. cDNA was diluted

1:100, qPCR reaction was assembled using Fast SYBR Green Mastermix (Applied Biosystem, Cat. No 4385612) and ran on a Lightcycler 480 II (Roche) instrument. Primer efficiencies were calculated via serial dilutions. Sequences as available in *Supplementary file 4*.

## Nuclear fractionation and western blot

For nuclear fractionation, 5–10 million cells were pelleted at 500 g for 5 min and washed with PBS. Cell pellets were either used directly or flash-frozen for later processing. Pellets were suspended in 300 μL of ice-cold NBT-10 buffer and rotated for 10 min at 4°C. Lysed cells were gently overlaid on 500 μL of ice-cold NB-1.2 buffer and spun at 5000 g for 20 min. Pelleted nuclei were washed once with 500 μL of ice-cold NB-10 buffer. Nuclei were resuspended in 60 μL of 1:1 mix of Protein-RIPA buffer and 5X Laemmli Sample buffer. Samples were boiled for 5 min at 95°C and ran on pre-cast 8% or 14% polyacrylamide gels (Serva, Cat. No 43260.01 and 43269.01) under denaturing conditions in 1X Running Buffer (25 mM Tris, 192 mM glycine, 0.1% SDS). Gels were wet-transferred onto nitrocellulose membranes (GE Healthcare Life Sciences, Cat. No 10600002) in ice-cold 1X Transfer Buffer (25 mM Tris, 192 mM glycine) with 20% methanol (for histones) or with 10% methanol + 0.1% SDS (for DOM proteins) at 400 mA for 45–60 min. Membranes were blocked with 5% BSA in TBS buffer for 1 hr and incubated with primary antibodies in TBST buffer (TBS + 0.1% Tween-20) + 5% non-fat milk at 4°C overnight. Membrane were washed 3 times (5 min each) with TBST buffer and incubated with secondary antibodies in TBST buffer for 1 hr at room temperature. Membrane were washed 3 times with TBST buffer, 2 times with TBS buffer (5 min each), and dried before imaging. Images were taken on a LI-COR Odyssey or a LI-COR Odyssey CLx machine (LI-COR Biosciences).

## ChIPseq

70 to 130 million *D. melanogaster* Kc167 cells, resuspended in 20 ml of complete Schneider's *Drosophila* Medium, were crosslinked by adding 1:10 of the volume of Fixing Solution (100 mM NaCl, 50 mM Hepes pH 8, 1 mM EDTA, 0.5 mM EGTA, 10% methanol-free formaldehyde) and rotated at room temperature for 8 min. 1.17 ml of freshly-prepared 2.5 M glycine was added to stop the fixation (final conc. 125 mM). Cells were pelleted at 500 g for 10 min (4°C) and resuspended in 10 mL of ice-cold PBS. 3.5 million of fixed *D. virilis* cells (fixed as described for *D. melanogaster* cells) were added for every 70 million *D. melanogaster* cells. Cells were pelleted at 526 g for 10 min (4°C) and resuspended in 1 ml of PBS + 0.5% Triton-X-100 + 1X cOmplete EDTA-free Protease Inhibitor for every 70 million *D. melanogaster* cells and rotated at 4°C for 15 min to release nuclei. Nuclei were pelleted at 2000 g for 10 min and washed once with 10 ml of ice-cold PBS. Nuclei were suspended in 1 ml of RIPA buffer (10 mM Tris-HCl pH 8, 140 mM NaCl, 1 mM EDTA, 0.1% Na-deoxycholate, 1% Triton-X-100, 0.1% SDS, 1 mM PMSF, 1X cOmplete EDTA-free Protease Inhibitor) + 2 mM $CaCl_2$ for every 70 million *D. melanogaster* cells, divided into 1 ml aliquots and flash-frozen in liquid $N_2$. 1 mL of fixed nuclei was quickly thawed and 1 μL of MNase (to 0.6 units) (Sigma-Aldrich, Cat. No N5386) added. Chromatin was digested for 35 min at 37°C. MNase digestion was stopped by transferring the samples on ice and adding 22 μL of 0.5 M EGTA. Samples were mildly sonicated using a Covaris S220 instrument with the following settings: 50 W peak power, 20% duty factor, 200 cycles/burst, 8 min total time. Insoluble chromatin was removed by centrifugation at 16,000 g for 30 min at 4°C. Soluble chromatin was pre-cleared by incubation with 10 μL of 50% RIPA-equilibrated Protein A + G sepharose bead slurry (GE Healthcare, Cat. No 17-5280-11 and 17-0618-06) for every 100 μL of chromatin for 1 hr at 4°C. 100 μL of pre-cleared chromatin were set aside (input) and kept overnight at 4°C, while each primary antibody was added to 300 μL of chromatin and incubated overnight at 4°C. 40 μL of 50% RIPA-equilibrated Protein A + G sepharose bead slurry was added for each immunoprecipitation and rotated 3 hr at 4°C. Beads were washed 5 times with 1 ml of RIPA (5 min rotation at 4°C, pelleted at 3000 g for 1 min between washes) and resuspended in 100 μL of TE (10 mM Tris pH 8, 1 mM EDTA). 0.5 μL of RNAseA (Sigma-Aldrich, Cat. No. R4875) was added to both input samples and resuspended beads, followed by incubation at 37°C for 30 min. After addition of 6 μL of 10% SDS, protease digestion (250 ng/ μL Proteinase K, Genaxxon, Cat.no. M3036.0100) and crosslink reversal were performed simultaneously at 68°C for 2 hr. DNA was purified using 1.8X Agencourt AMPure XP beads (Beckman Coulter, Cat No A63880) following the standard protocol and eluted in 30 μL of 5 mM Tris-HCl pH 8. Libraries for sequencing were prepared using NEBNext Ultra II DNA library prep kit for Illumina (New England Biolabs, E7465). Libraries

were sequenced on an Illumina HiSeq 1500 instrument at the Laboratory of Functional Genomic Analysis (LAFUGA, Gene Center Munich, LMU). 50 bp single-end reads were mapped to the *D. melanogaster* (release 6) or independently to the *D. virilis* (release 1) genome using *bowtie2* (*Langmead and Salzberg, 2012*) using standard parameters. Tag directories and input-normalized coverage files were generated using *Homer* (*Heinz et al., 2010*) with the parameters `-fragLength 150` and `-totalReads` was set to the number of reads mapped to *D. virilis* genome. Input-normalized, scaled *D. melanogaster* coverage files were visualized using the Integrative Genomics Viewer (*Robinson et al., 2011*). Scripts for *D. virilis* scaling and input normalization are available on GitHub (https://github.com/tschauer/Domino_ChIPseq_2020). Composite plots were generated using *tsTools* (https://github.com/musikutiv/tsTools) and base R graphics. Annotations are derived from *TxDb.Dmelanogaster.UCSC.dm6.ensGene_3.4.4* (http://bioconductor.org/packages/release/data/annotation/html/TxDb.Dmelanogaster.UCSC.dm6.ensGene.html). Heatmaps were generated using *pheatmap* (https://cran.r-project.org/web/packages/pheatmap/index.html). Violin-boxplots were generated using *ggplot2* (https://cran.r-project.org/web/packages/ggplot2/index.html).

## Cloning of DOM constructs

DOM-A and DOM-B cDNAs were cloned into pENTR3c vector (Thermo Fischer Scientific, Cat. No A10464) by In-Fusion Cloning (Takara Bio, Cat. No 638909) using LD35056, LD03212, LD32234 plasmids *Drosophila* Genomic Resource Center) as templates for PCR. RNAi-resistant DOM-A and DOM-B were generated by substituting around 500 bp of the original cDNA sequence with manually mutagenized, synthesized DNA constructs (gBlock, Integrated DNA Technology), by restriction cloning. DOM-A and DOM-B K945G mutants were generated by site-directed mutagenesis (New England Biolabs, Cat. No E0554S). For transfection in *Drosophila* cells, the constructs in pENTR3c were recombined in pHWG (expression driven by *hsp70* promoter, C-terminal GFP tag; *Drosophila* Genomic Resource Center) by Gateway cloning (Thermo Fischer Scientific).

## Complementation assays and immunofluorescence

1–2 million Kc167 cells in 2 ml complete Schneider's *Drosophila* Medium were seeded in each well of a 6-well plate. After 4 hr, cells were transfected with 500 ng of complementation plasmid (described before) + 25 ng of pCoBlast (Thermo Fischer, Cat. No K5150-01) using Effectene Transfection Reagent (QIAgen, Cat. No 301425). 48 hr after transfection, 2 ml of the cells were collected, transferred into T-25 flasks and diluted with 4 ml of complete Schneider's *Drosophila* Medium + Blasticidin at a final concentration of 50 ng/ul. 7–8 days after selection the cells were collected and treated with dsRNA as described before.

For immunofluorescence, 0.2–0.4 million cells in 200 µL of complete Schneider's *Drosophila* Medium were seeded onto a round 12 mm coverslips (Paul Marienfeld GmbH and Co., Cat No. 0117520) placed separately inside wells of 12-well plates. Cell were allowed to attach for 2–4 hr and the coverslips were gently rinsed with 500 µL of PBS. Cells were fixed in 500 µL of ice-cold PBS + 2% formaldehyde for 7.5 min. After removal of fixative, cells were permeabilized by adding 500 µL of ice-cold PBS + 0.25% Triton-X-100 + 1% formaldehyde and incubating for 7.5 min. Coverslips were washed two times with 1 ml of PBS and blocked with PBS + 3% BSA for 1 hr at room temperature. Coverslip were transferred onto a piece of parafilm, placed into a wet chamber and 40 µL of primary antibody solution was gently added onto the coverslip. After overnight incubation at 4°C, coverslips were transferred back to 12-well plates and washed twice with 1 ml of PBS. Coverslip were transferred again onto a piece of parafilm, placed into a wet chamber and 40 µL of secondary antibody was gently added onto the coverslip. After 1 hr incubation at room temperature, coverslips were transferred back to 12-well plates and washed twice with 1 ml of PBS. Cells were incubated with 1 ml of 0.2 µg/ml DAPI (Sigma-Aldrich, Cat. No 10236276001) for 5 min at room temperature. Coverslips were washed with PBS and with deionized water, mounted on slides with 8 µL of Vectashield (Vector Laboratories, Cat. No H-1000) and sealed with nail polish. Images were taken on a Leica SP5 confocal microscope. Images were processed and analyzed using Fiji (*Source code 2*) and data plotted using R-Studio. p-values were calculated using linear regression (*lm* function in R).

## Histone extraction and targeted mass-spectrometry

Kc167 cells were treated with dsRNAs in 6-well plates as described before. Cells were counted, pelleted and snap-frozen in liquid $N_2$. For histone acid extraction, pellets from 4 to 12 million cells were resuspended in 500 µL of ice-cold 0.2M $H_2SO_4$ and histone were extracted by rotating overnight at 4°C. Cell debris were removed by centrifugation at 16,000 g for 10 min at 4°C. Histone were precipitated by adding trichloroacetic acid to reach 26% final concentration. Tubes were mixed and incubated at 4°C for 2 hr and spun at 16,000 g for 45 min. Pellets were washed twice with ice-cold 100% acetone (5 min rotation at 4°C, 10 min of 16,000 g spin at 4°C between washes), dried for 30 min at room temperature and resuspended in 10 µL of 2.5x Laemmli sample buffer for every initial cell million and boiled at 95°C for 5 min. Samples were stored at −20°C until further use. The histones corresponding to 10 million cells were separated onto 4–20% pre-cast polyacrylamide gels (Serva, Cat. No 43277.01). Gels were briefly stained with Coomassie (Serva, Cat. No 17524.01) and stored in water at 4°C. For targeted mass-spectrometry analysis, histones were excised, washed once with water and de-stained twice by incubating 30 min at 37°C with 200 µL of 50% acetonitrile (ACN) in 50 mM $NH_4HCO_3$. Gel pieces were then washed twice with 200 µL water and twice with 200 µL of 100% ACN to dehydrate them, followed by 5 min of speed-vac to remove residual ACN. Histones were in-gel acylated by adding 10 µL of deuterated acetic anhydride (Sigma-Aldrich, Cat. No 175641) and 20 µL of 100 mM $NH_4HCO_3$. After 1 min, 70 µL of 1 M $NH_4HCO_3$ were slowly added to the reaction. Samples were incubated at 37°C for 45 min with vigorous shaking. Samples were washed 5 times with 200 µL of 100 mM $NH_4HCO_3$, 5 times with 200 µL of water and twice with 200 µL of 100% ACN, followed by 3 min of speed-vac. Gel pieces were rehydrated in 20 µL of trypsin solution (25 ng/ µL trypsin in 100 mM $NH_4HCO_3$) (Promega, Cat. No V5111) and incubated at 4°C for 20 min. After the addition of 100 µL of 50 mM $NH_4HCO_3$, histones were in-gel digested overnight at 37°C. Peptides were sequentially extracted by incubating 10 min at room temperature twice with 60 µL of 50% ACN 0.25% trifluoroacetic acid (TFA) and twice 40 µL of 100% ACN. The total volume (around 250 µL) was evaporated in a centrifugal evaporator and the dried peptides were stored at −20°C until resuspension in 100 µL of 0.1% TFA. Peptides were loaded in a C18 Stagetip (pre-washed with ACN and conditioned with 0.1% TFA), washed 3 times with 20 µL of 0.1% TFA and peptides were eluted 3 times with 20 µL of 80% ACN 0.25% TFA. Eluted peptides were evaporated in a centrifugal evaporator, resuspended in 15 µl of 0.1% TFA and stored at −20°C. Desalted peptides were injected in an RSLCnano system (Thermo Fisher Scientific) and separated in a 15 cm analytical column (75 µm ID home-packed with ReproSil-Pur C18-AQ 2.4 µm from Dr. Maisch) with a 50 min gradient from 4% to 40% acetonitrile in 0.1% formic acid at 300 nl/min flowrate. The effluent from the HPLC was electrosprayed into Q Exactive HF mass spectrometer (Thermo Fisher Scientific). The MS instrument was programmed to target several ions as described before (*Feller et al., 2015*) except for the MS3 fragmentation. Survey full scan MS spectra (from m/z 270–730) were acquired with resolution R = 60,000 at m/z 400 (AGC target of $3 \times 10^6$). Targeted ions were isolated with an isolation window of 0.7 m/z to a target value of $2 \times 10^5$ and fragmented at 27% normalized collision energy. Typical mass spectrometric conditions were: spray voltage, 1.5 kV; no sheath and auxiliary gas flow; heated capillary temperature, 250°C. Peak integration was performed using Skyline (https://skyline.ms/project/home/software/Skyline/begin.view). Quantified data was further analyzed in R according to the formulas described in *Feller et al. (2015)* (*Supplementary file 5*; *Source code 3* and *Source code 4*).

## Antibodies

DOM-A and DOM-B polyclonal antibody were generated by expression of C-terminal specific polypeptides. For DOM-A, residues 2963 to 3188 were expressed as C-terminal Glutathione-S-transferase (GST) fusion in *E. coli*, purified using Glutathione Sepharose resin (GE Healthcare, Cat. No 17075605) and eluted with glutathione. For DOM-B, residues 2395 to 2497 were expressed as C-terminal Maltose Binding Protein (MBP) fusion in *E. coli*, absorbed to amylose resin (New England Biolabs, Cat. No E8121S) and eluted with maltose. Antibody production in rabbit was done by Eurogentec (https://secure.eurogentec.com/eu-home.html). Both antibodies were validated by RNAi and western blot. For the monoclonal antibody against TIP60, N-terminal 6xHis-tagged TIP60 (full length) was expressed in *E. coli*, purified over a Ni-NTA column and eluted with imidazole.

Monoclonal antibodies were developed by Dr. Elizabeth Kremmer (BioSysM, LMU Munich). Antibodies were validated by RNAi and western blot.

Sources of other antibodies were: DOM-A monoclonal and H2A.V polyclonal (*Börner and Becker, 2016*). Histone H4 rabbit polyclonal antibody: Abcam (Cat. No ab10158). Mouse anti- FLAG monoclonal antibody: Sigma-Aldrich (Cat. No F3165). Anti H4K12ac rabbit polyclonal antibody: Merck-Millipore (Cat. No 07–595). Anti-GFP mouse monoclonal antibody: Roche (Cat. No 11814460001). Anti-lamin mouse monoclonal antibody: kind gift of Dr. Harald Saumweber.

### Data and code availability

Next Generation sequencing data are available at the Gene Expression Omnibus under accession number GSE145738.

Targeted proteomics data are available at ProteomeXchange under accession number PXD017729.

Scripts for *D. virilis* scaling and input normalization for ChIP-seq are available on GitHub (https://github.com/tschauer/Domino_ChIPseq_2020; *Schauer, 2020a*; copy archived at https://github.com/elifesciences-publications/Domino_ChIPseq_2020).

Scripts for RNA-seq analysis are available on GitHub (https://github.com/tschauer/Domino_RNA-seq_2020; *Schauer, 2020b*; copy archived at https://github.com/elifesciences-publications/Domino_RNAseq_2020).

Immunofluorescence images used for quantification of the complementation assays are available on Dryad (https://doi.org/10.5061/dryad.1rn8pk0qt).

## Acknowledgements

This work was supported by the Deutsche Forschungsgemeinschaft (DFG, German Research Council) – Project ID 213249687 - SFB1064-A1 and -Z04. ZA was supported by the EMBO long-term fellowship (ALTF 168–2018). PH was supported by a Wellcome Trust Senior Fellowship award (103897) and the Wellcome Centre for Cell Biology is supported by core funding from the Wellcome Trust (092076). We thank K Förstemann, J Müller and the Becker laboratory for discussion; R Villa, M Müller and S Baldi for useful feedback and critical reading of the manuscript; H Blum and S Krebs for the next-generation sequencing; I Fornè, AV Venkatasubramani and A Imhof for the targeted mass-spectrometry; M Prestel for the initial generation of the TIP60 antibody and N Steffen and K Börner for initial characterization.

## Additional information

### Funding

| Funder | Grant reference number | Author |
|---|---|---|
| Deutsche Forschungsgemeinschaft | SFB1064-A1 | Alessandro Scacchetti<br>Aline Campos Sparr<br>Silke Krause<br>Peter B Becker |
| Deutsche Forschungsgemeinschaft | SFB1064-Z04 | Tamas Schauer |
| Wellcome | 103897 | Patrick Heun |
| EMBO | ALTF 168-2018 | Zivkos Apostolou |

The funders had no role in study design, data collection and interpretation, or the decision to submit the work for publication.

### Author contributions

Alessandro Scacchetti, Conceptualization, Data curation, Formal analysis, Investigation, Visualization, Methodology, Writing - original draft, Writing - review and editing; Tamas Schauer, Data curation, Software, Formal analysis, Visualization; Alexander Reim, Data curation, Formal analysis, Investigation, Methodology; Zivkos Apostolou, Aline Campos Sparr, Silke Krause, Investigation; Patrick Heun,

Supervision, Methodology, Supervised the CRISPR/Cas9 tagging; Michael Wierer, Supervision; Peter B Becker, Conceptualization, Supervision, Funding acquisition, Writing - original draft, Project administration, Writing - review and editing

### Author ORCIDs
Alessandro Scacchetti  https://orcid.org/0000-0002-0254-3717
Patrick Heun  http://orcid.org/0000-0001-8400-1892
Peter B Becker  https://orcid.org/0000-0001-7186-0372

### Decision letter and Author response
Decision letter https://doi.org/10.7554/eLife.56325.sa1
Author response https://doi.org/10.7554/eLife.56325.sa2

## Additional files
### Supplementary files
• Source code 1. R script for analysis of AP-MS data.

• Source code 2. Java script for quantification of immunofluorescence pictures.

• Source code 3. R script for quantification of acetylated peptides.

• Source code 4. R script for analysis of acetylated peptides.

• Supplementary file 1. Excel spreadsheet containing imputed LFQ values obtained from the MaxLFQ algorithm, *limma* output and DOM-A or DOM-B specific interactors.

• Supplementary file 2. Excel spreadsheet containing result tables from DEseq2 analysis.

• Supplementary file 3. Comparison of the known subunits of SWR1- and NuA4-type complexes between *D. melanogaster*, *S. cerevisiae* and *H. sapiens*. Subunit composition of the yeast SWR1 and NuA4 were obtained from the manually-curated SGD database (https://www.yeastgenome.org) (CPX-2122 and CPX3155). For the human complexes, we refer to the EP400 complex subunits described in *Dalvai et al., 2015* and to the SRCAP subunits described in *Feng et al., 2018*.

• Supplementary file 4. gRNAs, repair templates and primers used in this study.

• Supplementary file 5. Excel spreadsheet containing raw output from Skyline analysis and results from quantification of acetylated peptides.

• Transparent reporting form

### Data availability
Sequencing data have been deposited in GEO under accession code GSE145738. Targeted proteomics data are available at ProteomeXchange under accession number PXD017729. Immunofluoresce images are available at Dryad under accession number https://doi.org/10.5061/dryad.1rn8pk0qt.

The following datasets were generated:

| Author(s) | Year | Dataset title | Dataset URL | Database and Identifier |
|---|---|---|---|---|
| Scacchetti A, Schauer TR, Apostolou Z, Sparr AC, Krause S, Heun P, | 2020 | Drosophila SWR1 and NuA4 complexes originate from DOMINO splice isoforms | https://www.ncbi.nlm.nih.gov/geo/query/acc.cgi?acc=GSE145738 | NCBI Gene Expression Omnibus, GSE145738 |

| | | | | |
|---|---|---|---|---|
| Wierer M, Becker PB | | | | |
| Scacchetti A, Schauer TR, Apostolou Z, Sparr AC, Krause S, Heun P, Wierer M, Becker PB | 2020 | Drosophila SWR1 and NuA4 complexes are defined by DOMINO isoform | http://proteomecentral.proteomexchange.org/cgi/GetDataset?ID=PXD017729 | ProteomeXchange, PXD017729 |
| Scacchetti A, Schauer TR, Apostolou Z, Sparr AC, Krause S, Heun P, Wierer M, Becker PB | 2020 | Drosophila SWR1 and NuA4 complexes are defined by DOMINO isoform | https://doi.org/10.5061/dryad.1rn8pk0qt | Dryad Digital Repository, 10.5061/dryad.1rn8pk0qt |

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
