## [Decision Letter]

**Acceptance summary:**

While mammals contain two versions of the NuA4/TIP60 histone acetylase/ATPase complexes defined by paralogs of the ATPase subunit, this study shows that in flies and perhaps other insects this specialization is achieved by alternative splicing of the Domino ATPase. One version of the Domino complex is primarily involved in histone acetylation whereas the other is primarily involved in exchange of the histone variant H2Av.

**Decision letter after peer review:**

Thank you for submitting your article "*Drosophila* SWR1 and NuA4 complexes are defined by DOMINO isoforms" for consideration by *eLife*. Your article has been reviewed by three peer reviewers, including Jerry L Workman as the Reviewing Editor and Reviewer #1, and the evaluation has been overseen by Kevin Struhl as the Senior Editor. The following individual involved in review of your submission has agreed to reveal their identity: Vikki Weake (Reviewer #2).

The reviewers have discussed the reviews with one another and the Reviewing Editor has drafted this decision to help you prepare a revised submission.

Summary:

The manuscript entitled "*Drosophila* SWR1 and NuA4 complexes are defined by DOMINO isoforms" by Scacchetti et al. demonstrates that the two alternatively spliced products of *domino* (*dom*) form two distinct complexes: DOM-A.C and DOM-B.C. They further show that DOM-A.C resembles the human EP400 complex, which seems to be a joint complex containing the yeast NuA4.C and some components of the SWR1.C. This DOM-A.C contains TIP60 and may target H4K12 and K5, and its HAT activity does not depend on the ATPase activity of DOM-A. On the other hand, the DOM-B.C is more like the conventional SWR1 complex and is responsible for putting H2Av onto the chromatin. This manuscript clearly demonstrates an interesting example in evolution that, while mammals separate the functions of SWR1.C and NuA4.C by gene duplication, flies achieve the same result by alternative splicing. However, some clarifications and improvement of either statements or experimental data (ex. ChIP-seq) are needed as detailed below.

Essential revisions:

1) Figure 1—figure supplement 1A: Why is the size of the band of the DOM-A clone #2 different from those of the other two clones? According to the legend, they should all be 72 bp longer than in the untagged cells.

2) Figure 1C: The authors described that Mass-spectrometry revealed 13 and 12 interactors for DOM-A and DOM-B, respectively. And they said, "of those, 8 are common between the two isoforms…" However, only 7 common interactors can be found in Figure 1C: BRD8, DMAP1, GAS41, MRGBP, BAP55, YL-1 and MRG15.

3) Figure 1D: REPT, PONT and HCF seem to be missing in this figure. If the authors do not define REPT and PONT as specific interactors of DOM-B (as they are shown in yellow in Figure 1C; DOM-A), please clarify whether HCF is a specific interactor or not by putting it in Figure 1C (DOM-A) or in Figure 1D.

4) Figure 1E: It is very difficult to understand that anti-DOM-A showed a band in unbound with DOM-A IP but no band in unbound with DOM-B IP (the DOM-A levels are similar in input). Also, since anti-DOM-A and anti-DOM-B can detect a band in unbound with DOM-A IP, it is awkward that there were no bands in unbound with no tag (input showed similar levels).

5) Figure 2: The main conclusion of this manuscript is that the DOM-A.C regulates transcription by acetylating H4K12 and the DOM-B.C regulates transcription by incorporating H2Av genome-wide. However, in Figure 2B, while the transcriptome profiles from cells treated with dsRNAs against DOM-A or TIP60 are closely positioned, those from cells treated with dsRNAs against DOM-B or H2Av are far apart (PC1/PC2). Furthermore, the r values of DOM-B data sets compared with DOM-A or H2Av are 0.45 (Figure 2C) or 0.51 (Figure 2D), respectively, which are pretty similar. If the transcriptional responses upon loss of DOM-A or DOM-B are "clearly different", then those upon loss of DOM-B or H2Av are also clearly different.

The authors tried to explain this discrepancy in Discussion: "DOM-B.C may also impact transcription independent of H2Av incorporation, as has been found for SWR1 (Morillo-Huesca et al., 2010)." However, this cited reference actually demonstrated that the genome-wide transcriptional misregulation in htz1d strain can be rescued by inactivation or deletion of SWR1. This result suggests that "SWR1 affects chromatin integrity because of an attempt to replace H2A with Htz1 in the absence of the latter", which further connects the function of SWR1 with Htz1. Therefore, the authors need to provide a more solid explanation for the discrepancy between the transcriptional responses upon loss of DOM-B or H2Av, in order to claim the main conclusion of this manuscript.

6) Figure 3—figure supplement 1C: An RT-qPCR control to show the knockdown efficiencies of all DOM-A.C or DOM-B.C components is required to claim "only RNAi against the DOM-B.C-specific subunit ARP6 reduced H2Av levels to a similar extent."

7) Figure 4 and Figure 4—figure supplement 1: Based on the labeling in Figure 4—figure supplement 1C-D, it seems that the B_1 sample was omitted from the analysis. The reason to omit this sample cannot be found in the Materials and methods. Please provide the reason.

8) Figure 4—figure supplement 1B: Upon TIP60 knock-down, while 1338 genes were down-regulated (Figure 2—figure supplement 1B), the H4K12ac and H4K5ac levels were only reduced 29% and 23%, respectively. Is H4K12ac levels reduced specifically at those 1338 genes? Similarly, this question also applies to DOM-A knock-down. The correlation between the down-regulated genes and the sites with reduced H4K12ac occupancies upon TIP60 or DOM-A knock-down is needed.

9) Figure 4—figure supplement 1C-F: The changes in H3K14ac and H3K23ac upon DOM-A or TIP60 RNAi are not very consistent between replicates (A_1 is even outside the clustering and B_3 is within two replicates of TIP60). Also, the text regarding Figure 4—figure supplement 1C-F are all descriptive without any interpretation about the possible biological meaning of these results.

10) Figure 4B and Figure 4—figure supplement 1G: The replicates of TIP60 and DOM-A RNAi are not consistent. For example, the middle track of TIP60 apparently shows more reads across the whole region (Figure 4B). Also, the middle track of DOM-A apparently shows less reads. More obviously, in Figure 4—figure supplement 1G, the correlation coefficients between TIP60_2 and _1/_1.2 are 0.58 and 0.4, which are very low as biological replicates. Even between TIP60_1 and _1.2, the coefficient is as low as 0.64. Also, the coefficients between A2 and A1/A1.2 are also too low (0.63). These data sets need to be improved.

11) Figure 4B, C: The authors propose that the increase in H4K12ac upon DOM-B RNAi may be due to absence of DOM-B allowing more DOM-A activity. Does it work through loss of H2Av incorporation? The authors may test their hypothesis by H4K12ac ChIP-qPCR in H2Av or DOM-B RNAi cells at some representative loci.

12) The authors mentioned that the model for H2Av exchange during DNA damage proposed in Kusch et al. should be re-visited because the purified complex may be a mixture of DOM-A.C, DOM-B.C and maybe dINO80.C. However, this early work did knock down TIP60, a specific component of DOM-A.C, and showed that the removal of phosphorylated H2Av is delayed after DNA damage. This result actually suggests a possible link between DOM-A.C and DOM-B.C under DNA damage stress. Otherwise, DOM-A.C itself may have activity to exchange H2Av upon DNA damage. The authors should test whether DOM-A.C and DOM-B.C functionally interact with each other upon DNA damage or DOM-A.C can perform H2Av exchange upon DNA damage.

---

## [Author Response]

Essential revisions:1) Figure 1—figure supplement 1A: Why is the size of the band of the DOM-A clone #2 different from those of the other two clones? According to the legend, they should all be 72 bp longer than in the untagged cells.

Indeed, the size of the amplicon of DOM-A clone #2 is larger than the other two DOM-A clones. We re-analyzed the edited locus by PCR and Sanger sequencing and discovered an insertion of 28 bp in the 3’ UTR of the DOM gene only in this particular clone, 29 bp downstream of the stop codon. The detailed sequence of the insertion is now shown in Figure 1—figure supplement 1B. We don’t know the nature and the origin of this insertion but, as both DOM-A protein expression, protein size (Figure 1B) and interactome (Supplementary file 1) in clone #2 don’t show major differences compared to the other clones, we consider this as a silent insertion.

2) Figure 1C: The authors described that Mass-spectrometry revealed 13 and 12 interactors for DOM-A and DOM-B, respectively. And they said, "of those, 8 are common between the two isoforms…" However, only 7 common interactors can be found in Figure 1C: BRD8, DMAP1, GAS41, MRGBP, BAP55, YL-1 and MRG15.

The reviewers are correct, the common interactors are indeed 7 and not 8. By mistake, we considered DOM itself as a common interactor. We now updated the text.

3) Figure 1D: REPT, PONT and HCF seem to be missing in this figure. If the authors do not define REPT and PONT as specific interactors of DOM-B (as they are shown in yellow in Figure 1C; DOM-A), please clarify whether HCF is a specific interactor or not by putting it in Figure 1C (DOM-A) or in Figure 1D.

We define as isoform-specific interactors those proteins that are significantly enriched (false discovery rate < 0.05) in the affinity purification of one or the other isoform when the pulldowns are compared (Figure 1D). No data point was excluded from the plot in Figure 1D, and the results of the statistical analysis are found in Supplementary file 1. PONT, REPT and HCF do not meet the significance criteria (FDR > 0.05) and are therefore not considered isoform-specific interactors. For clarity, HCF is now added to the figure.

4) Figure 1E: It is very difficult to understand that anti-DOM-A showed a band in unbound with DOM-A IP but no band in unbound with DOM-B IP (the DOM-A levels are similar in input). Also, since anti-DOM-A and anti-DOM-B can detect a band in unbound with DOM-A IP, it is awkward that there were no bands in unbound with no tag (input showed similar levels).

We thank the reviewers for spotting this inconsistency. We could not easily explain these results unless referring to loading mistakes. Therefore, we repeated the immunoprecipitation with two different sets of clones and following the exact procedure we used for mass spectrometry (including benzonase treatment). The results are shown in Figure 1E and Figure 1 —figure supplement 1C) both DOM-A and DOM-B bands could be detected in the unbound fraction of the no-tag control and 2) we could see a reduction in intensity in the unbound fraction in the corresponding IP samples (more for DOM-B, less for DOM-A). The main text was modified accordingly.

5) Figure 2: The main conclusion of this manuscript is that the DOM-A.C regulates transcription by acetylating H4K12 and the DOM-B.C regulates transcription by incorporating H2Av genome-wide. However, in Figure 2B, while the transcriptome profiles from cells treated with dsRNAs against DOM-A or TIP60 are closely positioned, those from cells treated with dsRNAs against DOM-B or H2Av are far apart (PC1/PC2). Furthermore, the r values of DOM-B data sets compared with DOM-A or H2Av are 0.45 (Figure 2C) or 0.51 (Figure 2D), respectively, which are pretty similar. If the transcriptional responses upon loss of DOM-A or DOM-B are "clearly different", then those upon loss of DOM-B or H2Av are also clearly different.The authors tried to explain this discrepancy in Discussion: "DOM-B.C may also impact transcription independent of H2Av incorporation, as has been found for SWR1 (Morillo-Huesca et al., 2010)." However, this cited reference actually demonstrated that the genome-wide transcriptional misregulation in htz1d strain can be rescued by inactivation or deletion of SWR1. This result suggests that "SWR1 affects chromatin integrity because of an attempt to replace H2A with Htz1 in the absence of the latter", which further connects the function of SWR1 with Htz1. Therefore, the authors need to provide a more solid explanation for the discrepancy between the transcriptional responses upon loss of DOM-B or H2Av, in order to claim the main conclusion of this manuscript.

The reviewers are right: correlation values of 0.51 and 0.45 are in a similar range. They show that the two compared data sets are different yet contain many co-regulated genes. When it comes to comparing the correlation of depletion profiles for either DOM-A or DOM-B with H2A.V, the difference is clear: The correlation of DOM-A with H2A.V is 0.25, while the correlation between DOM-B and H2A.V is 0.51. This substantiates our conclusion that many effects of DOM-B depletion can be explained by H2A.V depletion. The revised text now reads: “The effects of H2A.V depletion were better correlated to DOM-B (r=0.51) than to DOM-A knock-down (r=0.25) (Figure 2B, D). Obviously, many effects of DOM-B depletion may be explained by its H2A.V deposition function, but the ATPase also affects transcription through different routes”.

The correlation of DOM-A and DOM-B depletion transcriptomes of 0.45 shows that many genes are regulated by both proteins, but around 2500 genes are affected differently by DOM-A or DOM-B knockdown (see Figure 2 —figure supplement 1B, last panel). We revised the text to better describe the results, by adding the following sentences: “The correlation value of 0.45 indicates that many genes are regulated similarly by both ATPases, but a significant number of genes are also differentially affected based upon depletion of either DOM-A or DOM-B (Figure 2 —figure supplement 1B)”.

We removed the reference to the paper Morillo-Huesca et al. from the discussion since upon further inspection we found the data presented there can be interpreted in different ways and the reference does not clarify the issue. The revised text now reads: “This discrepancy could be explained in several ways. […] Third, the global increase of H4K12ac at promoters upon DOM-B knock-down might indirectly compensate for the loss of H2A.V at some specific genes”.

6) Figure 3—figure supplement 1C: An RT-qPCR control to show the knockdown efficiencies of all DOM-A.C or DOM-B.C components is required to claim "only RNAi against the DOM-B.C-specific subunit ARP6 reduced H2Av levels to a similar extent."

Results of qPCR analysis are now included in Figure 3 —figure supplement 1D. As shown in the figure, with the exception of *Nipped-A*, the mRNA of the other subunits is reduced efficiently without major differences between the targets.

7) Figure 4 and Figure 4—figure supplement 1: Based on the labeling in Figure 4—figure supplement 1C-D, it seems that the B_1 sample was omitted from the analysis. The reason to omit this sample cannot be found in the Materials and methods. Please provide the reason.

As described in the figure legend, only 3 replicates for DOM-B where analyzed and none of the replicates was excluded from the analysis. For clarity we re-numbered the samples from _1 to _3.

8) Figure 4—figure supplement 1B: Upon TIP60 knock-down, while 1338 genes were down-regulated (Figure 2—figure supplement 1B), the H4K12ac and H4K5ac levels were only reduced 29% and 23%, respectively. Is H4K12ac levels reduced specifically at those 1338 genes? SImilarly, this question also applies to DOM-A knock-down. The correlation between the down-regulated genes and the sites with reduced H4K12ac occupancies upon TIP60 or DOM-A knock-down is needed.

We correlated the basal H4K12ac signal (i.e. control RNAi) at promoters of genes that were significantly up- or down-regulated in the knockdown of DOM-A or TIP60, and the results can now be found in Figure 4 —figure supplement 1G. We observed that in both cases down-regulated genes are more enriched in H4K12ac at their promoters in steady state. We then looked at the log2 fold-change (TIP60 or DOM-A knock-down versus control) of H4K12ac signal at promoters of the same gene classes (Figure 4 —figure supplement 1H). For both TIP60 and DOM-A RNAi, H4K12ac is reduced globally and not specifically at the significantly mis-regulated genes (although there is a trend of stronger H4K12ac loss at promoters of downregulated genes).

These findings are now presented as Figure 4 —figure supplement 1G, H, I and discussed in the main text. In the Discussion we speculate that these genes might rely more on H4K12ac for their expression or be more sensitive to changes in acetylation.

9) Figure 4—figure supplement 1C-F: The changes in H3K14ac and H3K23ac upon DOM-A or TIP60 RNAi are not very consistent between replicates (A_1 is even outside the clustering and B_3 is within two replicates of TIP60). Also, the text regarding Figure 4—figure supplement 1C-F are all descriptive without any interpretation about the possible biological meaning of these results.

Given the variability and the fact that little is known about the biological functions of these modifications in *Drosophila*, we decided to remove the plot. Raw data, along with other modifications measured but not described, is still available in Supplementary file 5.

10) Figure 4B and Figure 4—figure supplement 1G: The replicates of TIP60 and DOM-A RNAi are not consistent. For example, the middle track of TIP60 apparently shows more reads across the whole region (Figure 4B). Also, the middle track of DOM-A apparently shows less reads. More obviously, in Figure 4—figure supplement 1G, the correlation coefficients between TIP60_2 and _1/_1.2 are 0.58 and 0.4, which are very low as biological replicates. Even between TIP60_1 and _1.2, the coefficient is as low as 0.64. Also, the coefficients between A2 and A1/A1.2 are also too low (0.63). These data sets need to be improved.

We agree with the reviewers that the quality of the dataset is borderline, because the H4K12ac ChIP is not very efficient. Unfortunately, this is the first profiling of this modification and so there is no published dataset for reference. While we controlled for most of the critical variables (e.g. knock-down efficiency, MNase digestion, spike-in amounts, etc.) we still observe high variability between replicates. In the case of TIP60 knock-down, despite variability, we still observe a significant global loss of H4K12ac. We now alerted the reader to these shortcomings in the text: “We found the H4K12ac signal reduced in many regions of the genome, including promoters and transcriptional termination sites, upon TIP60 RNAi and to a lesser extent upon DOM-A RNAi (Figure 4B,C, Figure 4 —figure supplement 1F,H,I), but the results suffer from variability probably due to a low ChIP efficiency of the H4K12ac antibody (Figure 4 —figure supplement 1F).”. Additionally, to reduce the noise of the data, we re-calculated TSS/TTS coverages and correlation coefficients using the expressed genes (N=10138) instead of all genes (N = 17549). The overall results are not changed but the correlation coefficients between biological replicates are a bit higher (Figure 4 —figure supplement 1F).

If you insist, we can add additional replicates, but due to the shut-down of our facilities this may take some time.

11) Figure 4B,C: The authors propose that the increase in H4K12ac upon DOM-B RNAi may be due to absence of DOM-B allowing more DOM-A activity. Does it work through loss of H2Av incorporation? The authors may test their hypothesis by H4K12ac ChIP-qPCR in H2Av or DOM-B RNAi cells at some representative loci.

We performed H4K12ac ChIPseq upon H2A.V knockdown, now included in Figure 4 and Figure 4 —figure supplement 1 and the text. As in the case of DOM-B RNAi, we observe an increase in H4K12ac signal at promoters when H2A.V is depleted – although the magnitude of the effect is lower. We speculate that DOM-B effects on H4K12ac might be, at least partially, dependent on its ability to incorporate H2A.V. We now provide a possible explanation in the Discussion: “The increase in H4K12ac at promoter observed upon DOM-B RNAi appears to be partially phenocopied by the knock-down of H2A.V. It is possible that H2A-containing nucleosomes are a better substrate for the DOM-A.C compared to the ones containing only H2A.V. The loss of the variant might therefore result in higher H4K12ac catalyzed by TIP60”.

12) The authors mentioned that the model for H2Av exchange during DNA damage proposed in Kusch et al. should be re-visited because the purified complex may be a mixture of DOM-A.C, DOM-B.C and maybe dINO80.C. However, this early work did knock down TIP60, a specific component of DOM-A.C, and showed that the removal of phosphorylated H2Av is delayed after DNA damage. This result actually suggests a possible link between DOM-A.C and DOM-B.C under DNA damage stress. Otherwise, DOM-A.C itself may have activity to exchange H2Av upon DNA damage. The authors should test whether DOM-A.C and DOM-B.C functionally interact with each other upon DNA damage or DOM-A.C can perform H2Av exchange upon DNA damage.

The purpose of our work was to define the molecular signatures of DOM-A and DOM-B complexes the context of transcription. The characterization of their role during DNA damage response/repair is of great interest and warrants a deep and thorough investigation. Given the complexity of DNA damage response, teasing apart the effects of the two DOM complexes and H2A.V on DNA repair is beyond the scope of this manuscript. Nevertheless, we think this is a necessary and natural continuation of our current work and we added the following sentence to the Discussion: “It will be interesting to define the role of each complex on the recognition and restoration of damaged chromatin, especially at the level of H2A.V remodeling and acetylation-based signaling”.